# Extending to New Domains without Visual and Textual Oracles

## Abstract

To avoid the high cost of collecting visual data from all test domains in the domain adaptation task, recent work takes advantage of the pre-trained large-scale vision language models and augment training data with only text descriptions (e.g.,"a photo/painting/sketch...") of each test domain. However, in many real-world applications, such text information of test domains is not always available in advance. Moreover, even if we can verbalize all test domains, it is laborious for existing work (Dunlap et al., 2023) to train a different augmentation network for each possible unseen domain, which suffers from time-inefficiency. To overcome these challenges, we benefit from the multimodal embedding space of a pre-trained vision-language model and propose to acquire *training-free* and *domain-invariant* augmentations with text descriptions of arbitrary crafted unseen domains, which *not* necessarily match test domains. Beyond achieving state-of-the-art results, compared with existing works that require trainable augmentation networks, our approach is also notably more time-efficient, and exhibits a more solid theoretical support. *Code will be publicly available.*

## 1 Introduction

Traditional computer vision models are trained based on the assumption that the training and test data are identically and independently distributed (i.i.d.). However, this assumption does not always hold in practical scenarios. Therefore, it is critical for models to generalize to unseen distributions. Significant progress has been made in domain generalization (DG), where the goal is to train a model on several different but related domains, enabling robust generalization to unseen test domains. However, acquiring image data from multiple domains is often necessary, and it can be costly due to the challenge of collecting images from every desired domain.

To address this challenge, independent of visual data, LADS (Dunlap et al., 2023) proposes augmenting the image embeddings in the source domain with textual descriptions of target domains (Fig. 1 left). While LADS is more cost-effective compared to traditional DG models that require images from other domains, it remains demanding because, in many practical scenarios, obtaining precise text descriptions of all potential test domains in advance is not always feasible. Furthermore, training a different augmentation network for each test domain is laborious. Motivated by this, without visual or textual information from test domains, we are curious to explore whether it is possible to learn a model that performs well in test domains with the help of text descriptions of crafted unseen domains efficiently? (Example prompts are available in Fig. 3)

Benefiting from the learned multimodal embedding space in pre-trained large-scale vision-language (VL) models (Radford et al., 2021; Jia et al., 2021), We achieve this goal by performing **T**ext-driven **E**mbedding **A**ugmentations in a training-free **M**anner, termed **TEAM**. Different from LADS, TEAM does not require either a textual oracle of test domains (i.e., accessing to textual information of test domains in advance) nor any trainable augmentation networks, meanwhile it is more computationally efficient with stronger theoretical support.

In particular, while LADS augments image embeddings from the source domain to the test domain(s) with given test domain descriptions, we explore the possibility of acquiring descriptions of arbitrary domains (*not* necessarily matching the test domain exactly) and performing data augmentation with them to train a robust classifier, referred as the *Text-driven Domain Generalization* problem. Specifically, we design two frameworks: (1) TEAM-*invariant*, the time-efficient one, where we

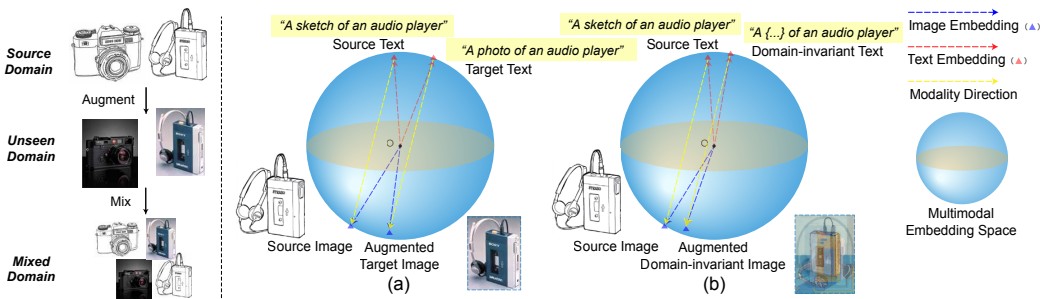

Figure 1: **Left**: Training pipeline of augmentation-based methods. Given source image data, they are first augmented to unseen domains with augmentation functions; then, the original and augmented data are mixed for training. Note that in our model, the augmentation happens in a latent space. **Right**: Two augmentation options. **(a)**: Given embeddings of the source image and text, and an unseen target domain text embedding, the target image embedding is obtained via a *training-free* augmentation by finding an embedding that aligns the two modality directions (yellow arrows). **(b)**: This option differs from (a) in that the unseen domain text embedding is replaced by a domain-invariant text embedding. Blue and red arrows represent image and text embeddings, respectively.

perform one-time augmentation to obtain domain-invariant augmentations for all acquired domain descriptions, and (2) TEAM-*full*, the time-consuming one, where we augment embeddings for each acquired domain description. Furthermore, we propose a new *training-free* embedding augmentation method based on the geometric characteristics of the embedding distribution in the multimodal embedding space. While LADS requires training different augmentation networks for *each* test domain, which is especially burdensome when there are numerous test domains, our method does not require any training for embedding augmentation. Not only is our method more time-efficient, but it is also more accurate, generalizable, and explainable. We summarize our contributions as follows:

- We explore an interesting yet under-explored problem, i.e., learning a model that extends well to test domains with only crafted text descriptions from arbitrary unseen domains *(not test domains)*. We call it Text-driven Domain Generalization problem.

- With the multimodal embedding space of a pre-trained VL model, we propose a novel training-free embedding augmentation method with theoretical guarantees, based on the geometric characteristics of the embedding distribution.

- Furthermore, combined with our training-free technique, we build a framework with our augmentation method that performs domain-invariant augmentations to solve the Text-driven Domain Generalization problem, which is more time-efficient while achieving better results than competing baselines.

## 2 RELATED WORK

**Contrastive Language-Image Pre-training.** CLIP (Radford et al., 2021) learns text-image matching from 400 million image-text pairs with contrastive learning, demonstrating strong zero-shot classification capability with well-structured multimodal embedding space. Benefiting from learnt knowledge in pre-trained CLIP, some recent works use it as the backbone for some specific tasks. In particular, linear probing is a popular technique towards this end (Kumar et al., 2022; Merullo et al., 2023; Dunlap et al., 2023), where a linear classifier is fit on the CLIP image or text embeddings.

**Domain Generalization.** The goal of classic domain generalization (DG) in image classification (Recht et al., 2019; Muandet et al., 2013; Li et al., 2017) is to learn a model from source domains, aiming to generalize well on unseen test domains. (Cho et al., 2023) work on source-free domain generalization, which particularly relies on the learnt general knowledge of common classes (e.g., dog, car, etc.) in CLIP, which however, cannot learn visual features of intricate classes from images. Free from the problem, LADS Dunlap et al. (2023) utilizes test domain descriptions for the augmentation of source images. In our problem, we have neither images nor text descriptions from test domains. Instead, our goal is to extend to unseen domains with a set of text descriptions of arbitrary unseen domains while preventing performance degradation on source domains.

**Image Augmentations with VL Models.** Owing to the great image-text matching capability of CLIP (Radford et al., 2021) and its well-structured multimodal embedding space. Recent methods (Gal et al., 2022; Patashnik et al., 2021; Ramesh et al., 2022) try to control the image generation in a generative model with language by leveraging CLIP. The multimodal embedding space makes it possible for them to embed an image into the multimodal space and edit the image embedding with text embeddings, which come from a natural language description of the intent augmentation. Different from these image editing techniques, (Sankaranarayanan et al., 2018) focuses on mitigating the domain gap via data augmentations with generative models, whose performances, however, are often limited by the quality of generated data. To overcome the bottleneck, LADS Dunlap et al. (2023) proposes to directly augment image embeddings in the CLIP latent space via trainable augmentation networks with text descriptions of test domains. Different from LADS, our method does not require either a textual oracle of test domains nor any trainable augmentation networks, meanwhile it is more computationally efficient with stronger theoretical support.

## 3 PRELIMINARY

We first recap LADS (Dunlap et al., 2023), then introduce the geometry and properties of the modality gap (Liang et al., 2022; Zhang et al., 2023) in multimodal embedding space.

### 3.1 RECAP OF LADS

**Problem Definition.** LADS (Dunlap et al., 2023) considers the problem of generalizing to test domains with source images and text descriptions of the test domains. Formally, we are given a training dataset $\{\boldsymbol{x}_i, y_i\}_{i=1}^n$ drawn from the source domain $D_{\text{training}}$, the class names $\text{t}_y$, a text description $t_{\text{source}}$ of the training domain, and a set of written descriptions $\{t_{\text{test}}^i\}_{i=1}^k$ of $k$ test domains $\{D_{\text{test}}^i\}_{i=1}^k$. The goal is to learn a model that works well on both source and test domains.

**Two-Stage Solution.** To solve the problem, LADS fits a linear classifier to the image embeddings of CLIP and performs a two-stage process: (1) training a set of augmentation networks $\{f_{\text{aug}}^i\}_{i=1}^k$ to augment source domain images for the corresponding $k$ test domains, and (2) training the linear probe (i.e., the classifier) on the source image embeddings and the augmented embeddings. Inference is performed by applying the linear probe to the CLIP image embeddings of the test images.

**Augmentation Network.** Let $h^I(\cdot)$ and $h^T(\cdot)$ denote the image and text encoder of CLIP, respectively. Let $(t_{\text{source}}; y_i)$ denote the composition of the domain description and the class name. For example, if $t_{\text{source}} = $ "*a sketch of a {}*" and $y_i = $ "*camera*", then $(t_{\text{source}}; y_i) = $ "*a sketch of a camera*". Given training point $(\boldsymbol{x}_i, y_i)$ and the text description $t_{\text{test}}^k$ of a test domain $k$, the overall loss for the training point consists of the domain alignment loss $\mathcal{L}_{\text{DA}}$ and the class consistency loss $\mathcal{L}_{\text{CC}}$. $\mathcal{L}_{\text{DA}}$ is based on the assumption that, there is a "global direction" that corresponds to a shift from $D_{\text{source}}$ to $D_{\text{test}}^k$ that is shared across both the image embedding space and text embeddings space. This "global direction" is defined as the normalized difference of the embeddings from the test domain and the embeddings from the source domain. This assumption originates from text-guided image generation (Gal et al., 2022; Patashnik et al., 2021). In particular:

$$\mathcal{L}_{\text{DA}}(f_{\text{aug}}^k) = \sum_{i=1}^n 1 - \left(\frac{f_{\text{aug}}^k(h^I(\boldsymbol{x_i})) - h^I(\boldsymbol{x_i})}{\left\|f_{\text{aug}}^k(h^I(\boldsymbol{x_i})) - h^I(\boldsymbol{x_i})\right\|} \cdot \frac{h^T(t_{\text{test}}^k; y_i) - h^T(t_{\text{source}}; y_i)}{\left\|h^T(t_{\text{test}}^k; y_i) - h^T(t_{\text{source}}; y_i)\right\|}\right), \quad (1)$$

$$\mathcal{L}_{\text{CC}}\left(f_{\text{aug}}^k\right) = \sum_{i=1}^n \text{Cross-entropy}\left(\text{Softmax}\left[f_{\text{aug}}^k\left(h^I(\boldsymbol{x_i})\right) \cdot h^T(y_i)\right], y_i\right), \quad (2)$$

where $\mathcal{L}_{\text{CC}}$ encourages the augmented embeddings to be distinguishable towards classes to keep their class information.

### 3.2 MODALITY GAP GEOMETRY

Recent research (Liang et al., 2022; Zhang et al., 2023) on the multimodal embedding space has revealed that, the learnt embeddings are approximately clustered per modality and there is a distinct modality gap, i.e., the distance between these clusters. Moreover, they empirically find several geometry characteristics: (1) The modality gap between corresponding image and text embeddings can be approximated by a constant vector, and (2) the modality gap is orthogonal to the span of image embeddings and text embeddings, and image embeddings and text embeddings have zero

mean in the subspace orthogonal to the modality gap. Motivated by above findings, we propose our training-free augmentation method.

## 4    TEXT-DRIVEN DOMAIN GENERALIZATION

In this section, we first introduce the problem definition, followed by a concise overview of our method. Subsequently, we delve into the details of our proposed training-free augmentation. Finally, we present the comprehensive framework of our method, TEAM.

**Problem Definition.** We address the challenge of generalizing to test domains using source images and text descriptions from various crafted domains. Formally, we are provided with a training dataset $\{x_i, y_i\}_{i=1}^n$ drawn from the source domain $D_{\text{training}}$, the class labels $t_y$, a text description $t_{\text{source}}$ of the source domain, and a set of text descriptions $\{t_{\text{crafted}}^i\}_{i=1}^k$ for $k$ crafted, previously unseen domains $\{D_{\text{crafted}}^i\}_{i=1}^k$ that are **_distinct_** from the test domains. Our objective is to develop a model that can generalize effectively to novel test domains.

**Pipeline.** As introduced in Fig. 1, given sources images, we first perform text-driven data augmentation (feature-level) with crafted unseen domain descriptions (i.e, target text) from a large language model. Then we train a linear probe on a mix of source and augmented image features.

### 4.1    TRAINING-FREE AUGMENTATION WITH MODALITY DIRECTION

We start from two questions on the augmentation stage: (1) is it necessary to train augmentation networks for each unseen target domain, which becomes more burdensome as the number of the descriptions of unseen domain increases? if not, how we can perform a training-free augmentation? and (2) is aligning the global direction in previous works (Dunlap et al., 2023; Gal et al., 2022; Patashnik et al., 2021) perfect for a training-free augmentation in terms of efficiency and accuracy?

**Global Direction**. The existence of global direction is based on the assumption that, in the CLIP embedding space, directions on two modalities that correspond to the same semantic changes to be roughly collinear. While it is effective for CLIP-guided image generation (Gal et al., 2022; Patashnik et al., 2021), we demonstrate that it is not flawless in our specific problem, especially when combined with our training-free augmentation.

*Training-free Augmentation*. To achieve training-free augmentation, we start from LADS and show that: given a training point $(x_i, y_i)$, regarding $f_{\text{aug}}^k(h^I(x_i))$ as a variable and let $\mathcal{L}_{DA} = 0$, we can have an analytical solution of it under moderate assumptions (detailed conclusion and proof are available in Appendix C). However, while the solution perfectly satisfies Eq. (1), it may not meet Eq. (2) (class consistency). To overcome the difficulties, motivated by recent findings on modality gap geometry (Zhang et al., 2023; Liang et al., 2022), we propose to align the *modality direction* to perform training-free augmentation, which proves to achieve higher performance with better theoretical supports.

**Modality Direction**. The modality direction is defined as the difference of embeddings from one modality to another. The yellow arrows in Fig. 1 and Fig. 2 visualize the modality directions of image-text pairs. Aligning modality direction is more appropriate than aligning global direction for training-free augmentation in following aspects: (1) better theoretical support, (2) better preservation of class information, and (3) milder assumption for an analytical solution.

*More Solid Theoretical Support*. Although the global direction is applied in a few recent works (Gal et al., 2022; Patashnik et al., 2021) on image generation, the assumption behind it is not validated. In contrast, (Zhang et al., 2023; Liang et al., 2022) empirically validate the existence of modality gap along with its geometry (Sec. 3.2), based on which we propose to align the modality direction.

*Better Preservation of Class Information*. While training augmentation networks with both Eq. (1) and Eq. (2) preserves domain and class information, aligning the global direction does not explicitly preserve class information since only Eq. (1) is satisfied. To address the challenge, we propose to align the modality direction, which naturally enhances the class consistency. Since the modality direction is approximately orthogonal to the span of embeddings (introduced in Sec. 3.2), the weight matrix of the learned classifier should also be approximately orthogonal to the modality direction. Hence the original prediction of the classifier is less affected if embeddings moves along the modality direction. In the example in Fig. 1(a), as the target text "*a photo of an audio player*"

is distinctive towards the class *audio player*, the (augmented) target image embedding, which is obtained by aligning the modality direction (the direction of modality change of the corresponding source data pair), should also be distinctive towards the class.

*Milder Assumption for an Analytical Solution.* We also show that, if aligning the global direction, the given pre-trained VL model should be perfect (i.e., the cosine similarity of the negative image-text pair is always less than that of the positive pair) to guarantee the existence of an analytical solution (Proposition 2 in Appendix C). However, this assumption can be relaxed if we align the modality direction (Proposition 1).

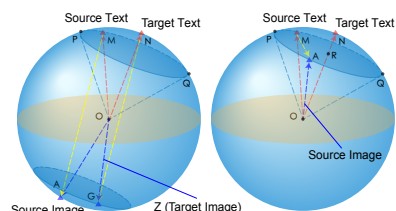

**Training-free Augmentation with Modality Direction**. Denote the output of the augmentation function $f_{\text{aug}}(\boldsymbol{x}, t_{\text{target}}; y, t_{\text{source}}; y)$ as a variable $\boldsymbol{z}$, by aligning the modality direction, the following equation should hold for each image-text pair $(\boldsymbol{x_i}, t_{\text{source}}; y_i)$:

$$\frac{\boldsymbol{z} - h^T(t_{\text{target}}; y_i)}{\|\boldsymbol{z} - h^T(t_{\text{target}}; y_i)\|} \cdot \frac{h^I(\boldsymbol{x_i}) - h^T(t_{\text{source}}; y_i)}{\|h^I(\boldsymbol{x_i}) - h^T(t_{\text{source}}; y_i)\|} = 1 \quad (3)$$

The embeddings are visualized in Fig. 2 (left). Instead of training an augmentation network $f_{\text{aug}}^k$ for each target domain $k$, and applying it to the original source image embedding $h^I(\boldsymbol{x_i})$ to obtain the augmented embedding as LADS does, we try to directly solve the Eq. (3) to obtain the corresponding target image embedding *without any trainable augmentation networks*, which is more explainable, time-efficient, lightweight yet more generalizable and more accurate.

Figure 2: Embeddings in feature space. **Left** (success case): Eq. (4) is satisfied, i.e., the target text embedding is closer to the source text embedding than the source image embedding, where Eq. (3) has a solution. **Right** (failure/boundary case): The circular surface $R$ is parallel to the tangent plane passing through point $N$. $\overrightarrow{ON} \cdot \overrightarrow{OM} = \overrightarrow{ON} \cdot \overrightarrow{OA}$, where Eq. (4) is not satisfied and Eq. (3) does not have a solution. Recent works Liang et al. (2022) and Zhang et al. (2023) validate that the distribution of CLIP (Radford et al., 2021) embeddings is consistent with the left figure.

**Lemma 1** *Given an image-text pair $(\boldsymbol{x_i}, t_{source}; y_i)$, the text description $(t_{target}; y_i)$ from target domain, and text encoder $h^T(\cdot)$ and image encoder $h^I(\cdot)$ from a pre-trained vision language model with a contrastive loss (e.g., the pre-trained CLIP). we have:*

$$h^T(t_{\text{target}}; y_i) \cdot h^T(t_{\text{source}}; y_i) > h^T(t_{\text{target}}; y_i) \cdot h^I(\boldsymbol{x_i}) \quad (4)$$

$$\text{subject to } ||h^I(\boldsymbol{x_i})|| = ||h^T(t_{\text{source}}; y_i)|| = ||h^T(t_{\text{target}}; y_i)|| \quad (5)$$

*Note that Eq. (5) is naturally satisfied as CLIP embeddings are normalized to a unit sphere.*

*Proof.* As stated in Sec. 3.2, in the multimodal space of CLIP: (1) embeddings are approximately clustered per modality, and (2) the modality gap between corresponding image and text embeddings can be approximated by a constant vector $\boldsymbol{g}$, which is orthogonal to the span of image embeddings and text embeddings. Let $h^I(\boldsymbol{x_i}) \approx \boldsymbol{g} + h^T(t_{\text{source}}; y_i)$, combining Eq. (5), we have $h^T(t_{\text{source}}; y_i) \cdot \boldsymbol{g} < 0$. As $\boldsymbol{g} \cdot (h^T(t_{\text{target}}; y_i) - h^T(t_{\text{source}}; y_i)) \approx 0$, we have $h^T(t_{\text{target}}; y_i) \cdot \boldsymbol{g} < 0$ (i.e., Eq. (4)) as well. Detailed proof is available in Appendix A. Fig. 2 (left) illustrates the embedding distribution.

**Proposition 1** *Given an image-text pair $(\boldsymbol{x_i}, t_{source}; y_i)$, the corresponding text description from target domain $(t_{target}; y_i)$, and text and image encoders $h^T(\cdot)$ and $h^I(\cdot)$ from a pre-trained vision language model, Eq. (3) has a solution if the vision-language model is pre-trained with a contrastive loss when subject to:*

$$||\boldsymbol{z}|| = ||h^I(\boldsymbol{x_i})|| = ||h^T(t_{\text{source}}; y_i)|| = ||h^T(t_{\text{target}}; y_i)|| \quad (6)$$

*Proof.* (Full proof is available in Appendix B) Consider Eq. 3, we have:

$$\boldsymbol{z} = \lambda(h^I(\boldsymbol{x_i}) - h^T(t_{\text{source}}; y_i)) + h^T(t_{\text{target}}; y_i), \quad (7)$$

where $\lambda$ is a non-negative coefficient. Combining with the constraint Eq. (6), the solution $\lambda$ and $z$ is derived:

$$\lambda = \frac{-2h^T(t_{\text{target}}; y_i) \cdot (h^I(\boldsymbol{x_i}) - h^T(t_{\text{source}}; y_i))}{(h^I(\boldsymbol{x_i}) - h^T(t_{\text{source}}; y_i))^2}, \quad (8)$$

$$\boldsymbol{z} = \frac{-2h^T(t_{\text{target}}; y_i) \cdot (h^I(\boldsymbol{x_i}) - h^T(t_{\text{source}}; y_i))}{(h^I(\boldsymbol{x_i}) - h^T(t_{\text{source}}; y_i))^2} \cdot (h^I(\boldsymbol{x_i}) - h^T(t_{\text{source}}; y_i)) + h^T(t_{\text{target}}; y_i) \quad (9)$$

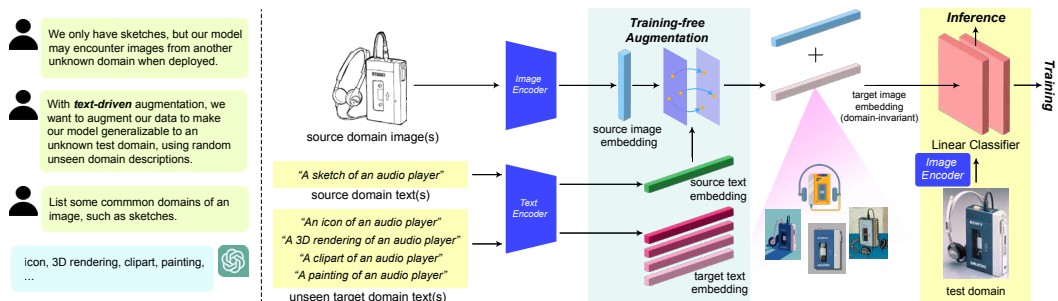

Figure 3: **TEAM-*invar.* framework**. Given source domain images (*sketch*), we first acquire several text descriptions of unseen domains (*different from the test domain*), and embed all texts and images into the CLIP embedding space. Then, a training-free augmentation is performed to obtain domain-invariant image embeddings under the guidance of crafted unseen domain descriptions. Finally, a linear classifier is trained on the mix of source embeddings and augmented embeddings.

Note that $z$ is the solution to Eq. (3) if and only if $\lambda > 0$, considering Eq. (8), i.e.,

$$-2h^T(t_{\text{target}}\,;y_i) \cdot (h^I(\boldsymbol{x_i}) - h^T(t_{\text{source}}\,;y_i)) > 0, \tag{10}$$

which is satisfied with a pre-trained vision-language model, e.g., CLIP (Lemma 1). So far we prove that Eq. (3) has an analytical solution under the constraint Eq. (6) with CLIP.

**Visual Explanation.** We also illustrate and prove that a solution to Eq. (3) can be guaranteed with a pre-trained CLIP from a geometrical perspective in Appendix B.1.

***Private* vs. *Global* Modality Direction.** In the above case, the modality direction is given by an image-text pair i.e., $(\boldsymbol{x_i}, t_{\text{source}}\,;y_i)$ in Eq. (3), which is referred as a private modality direction. We also propose to align the global modality direction as an alternative, which is given by all image-text pairs, instead of a single image-text pair. In this case, $h^I(\boldsymbol{x_i})$ and $h^T(t_{\text{source}};y_i)$ in Eq. (3) are replaced by $\mathbb{E}_{\boldsymbol{x}\sim\mathcal{X}}[h^I(\boldsymbol{x})]$ and $\mathbb{E}_{\boldsymbol{y}\sim\mathcal{Y}}[h^T(t_{\text{source}};y)]$. $\mathcal{X}$ and $\mathcal{Y}$ are the distribution of source domain images and labels, respectively. As we proved that a solution for Eq. (3) exists with a pre-trained CLIP for ***each*** image-text pair, by iterating all source domain image-text pairs over Eq. (10), we have: $-2h^T(t_{\text{target}}\,;y_i) \cdot (\mathbb{E}_{\boldsymbol{x}\sim\mathcal{X}}[h^I(\boldsymbol{x})] - \mathbb{E}_{\boldsymbol{y}\sim\mathcal{Y}}[h^T(t_{\text{source}};y)]) > 0$ for each target description $(t_{\text{target}}\,;y_i)$. It means when aligning the global modality direction with $\mathbb{E}_{\boldsymbol{x}\sim\mathcal{X}}[h^I(\boldsymbol{x})]$ and $\mathbb{E}_{\boldsymbol{y}\sim\mathcal{Y}}[h^T(t_{\text{source}};y)]$, a solution is still guaranteed. We refer the two methods on aligning private and global modality directions as TEAM (P) and TEAM (G) in later experiments, respectively.

### 4.2 GENERALIZING TO TEST DOMAINS WITH TEXT-DRIVEN EMBEDDING AUGMENTATION

**TEAM Framework.** Fig. 3 illustrates the overall TEAM-*invar.* framework. We have two stages. *(1) Training-free Augmentation.* Denote our augmentation function as $f_{\text{aug}}(\boldsymbol{x}, t_{\text{target}}; y, t_{\text{source}}; y)$, we either augment source domain image embeddings to each crafted unseen domain (Fig. 1 a): $E_{\text{aug}} = \{\{f_{\text{aug}}(\boldsymbol{x}_i, t^j_{\text{crafted}}; y_i, t_{\text{source}}; y_i)\}^n_{i=1}\}^k_{j=1}$, termed TEAM-*full*, or a domain-invariant space (Fig. 1 b): $E_{\text{aug}} = \{f_{\text{aug}}(\boldsymbol{x}_i, t_{\text{invar.}}; y_i, t_{\text{source}}; y_i)\}^n_{i=1}$, termed TEAM-*invar.* The latter is more time-efficient as the size of the augmented dataset $E_{\text{aug}}$ is $k$ times smaller, thus requires less training, while it is also informative because the domain-invariant representation contains rich domain-invariant class information. Next we discuss how to perform domain-invariant augmentation.

While it is intuitive to obtain a text description of a desired augmentation given a concrete domain name (e.g., if $t^k_{\text{crafted}}$ = "*a photo of a {}*" and $y_i$ = "*camera*", then $(t^k_{\text{crafted}}; y_i)$ = "*a photo of a camera*"), we are not able to have a literal description of an abstract "invariant" domain. To this end, we extract a domain-invariant text representation of a class with text representations of it from all crafted domains, then use it to guide the image embedding augmentation. Denote the extraction function as $g_{\text{extract}}(\cdot)$, we have $h^T(t_{\text{invar.}}; y_i) = g_{\text{extract}}(\{t^j_{\text{crafted}}; y_i\}^k_{j=1})$. Let $g_{\text{extract}}(\cdot) = $ Mean-Pooling$(\cdot)$, it would be: $\frac{\sum^k_{j=1}\{h^T(t^j_{\text{crafted}}; y_i)\}}{k}$. We discuss different choices of $g_{\text{extract}}(\cdot)$ later.

*(2) Training.* We train a linear probe on a mix of original image embeddings and augmented images embeddings $E_{\text{aug}}$. During inference, we apply the trained linear probe on the CLIP image embeddings of test images.

## 5 EXPERIMENTS

### 5.1 DATASETS

**CUB-Paintings.** It combines CUB-200 (Wah et al., 2011) and CUB-200-Paintings (Wang et al., 2020), where there are 200 different bird species from "photo" and "painting". **DomainNet.** Following (Dunlap et al., 2023), we use a specific split (Tan et al., 2020) of DomainNet (Peng et al., 2019) dataset which contains 40 most common classes from 4 domains: 'sketch', 'real', 'clipart', and 'painting'. Following prior works (Dunlap et al., 2023; Tan et al., 2020; Kumar et al., 2022), *we train on sketches and evaluate on the three other domains.* Details on experiment settings such as used crafted domain descriptions on each dataset is available in Appendix D.

### 5.2 BASELINES

*CLIP ZS generic (G)* only uses class name as the text craft (e.g. "camera"), while *CLIP ZS adaptive (A)* customizes text crafts for specific domains (e.g. "a painting of an airplane").

*CLIP LP* applies a linear classifier to the CLIP image embeddings. *CLIP LP (ZS init)* initializes the linear classifier with the text embeddings.

*WiSE-LP* (Wortsman et al., 2022) is an ensembling method which fine-tunes a CLIP model and does a weighted average of the fine-tuned model's weights with the original.

*VQGAN + CLIP* (Crowson et al., 2022) augment raw images in pixel space with a VQGAN (Esser et al., 2021) trained with CLIP. Following to LADS, due to the amount of time and compute required to generate images, the baseline only runs DomainNet.

*LADS* (Dunlap et al., 2023) is an augmentation-based method. Different from (Crowson et al., 2022), LADS leverages a pre-trained vision-language model to obtain text and image embeddings and performs the latent augmentation in the embedding space. Details on the choices of baselines and settings of them are available in Appendix D

### 5.3 IMPLEMENTATION DETAILS

Following Radford et al. (2021); Dunlap et al. (2023), all CLIP embeddings are normalized to a unit sphere. We use the official OpenAI CLIP model with a ViT-L backbone and resize images to 224x224. We run experiments on NVIDIA A100 GPUs.

For the LADS baseline, we adapt the default hyper parameters used in the original paper. For baselines that also appear in LADS, we directly show the results reported in LADS when applicable. In general we set the learning rate to 0.001, the weight decay to 0.05, and run experiments for 50 epochs. Full hyper parameter setting is available in Appendix D.

Following existing works (Dunlap et al., 2023), we report ID and OOD accuracy, and the average of the two. Note that the OOD performance is the major metric. We run each method over 3 different random seeds and report the mean and standard deviation.

### 5.4 RESULTS

**Extending to test domains *with* test domain oracle**. We first show the effectiveness of our data-free augmentation method by comparing TEAM with baselines *under the LADS setting*, where we have the test domain descriptions. Thus for all augmentation-based models, we directly augment the training data with given test domain descriptions. Tab. 1 illustrates the effectiveness of TEAM. Without trainable augmentation networks, we outperform VQGAN+CLIP Crowson et al. (2022) and LADS (Dunlap et al., 2023), which train augmentation networks for augmentation, in both ID and OOD (major metric).

We further compare TEAM (both TEAM-*full* and TEAM-*invar.*) with most competing baselines under our setting, where we *do not* have exact test domain descriptions. In this scenario, we propose to acquire several arbitrary crafted domain descriptions, with which we hope to train a classifier that generalizes well on test domains. In particular, we ask ChatGPT to list a number of domains, then we manually pick up some common domains that are different from test domains as crafted domains. Detailed choices are available in Appendix D.

**Extending to test domains *without* test domain oracle**. Tab. 2 shows experiment results under our text-drive domain generalization scenario. It is observed that the OOD performance gap be-

| Dataset | Method | Average | ID | OOD | Training-free Aug. |
|---|---|---|---|---|---|
| CUB-Paintings | CLIP ZS (G) | 56.59% | 60.34% | 52.84% | - |
| CUB-Paintings | CLIP ZS (A) | 58.16% | 61.93% | 54.38% | - |
| CUB-Paintings | CLIP LP | 75.12±0.18% | 85.91±0.08% | 64.33±0.29% | - |
| CUB-Paintings | CLIP LP (ZS init) | 75.57±0.06% | 86.08±0.11% | 65.05±0.05% | - |
| CUB-Paintings | WiSE-LP | 73.27±0.22% | 81.74±0.34% | 64.80±0.10% | - |
| CUB-Paintings | LADS | 76.16±0.23% | 86.14±0.29% | 66.18±0.25% | × |
| CUB-Paintings | TEAM (P) (Ours) | 76.76±0.25% | **86.41±0.21%** | 67.12±0.27% | ✓ |
| CUB-Paintings | TEAM (G) (Ours) | **76.94±0.21%** | 86.40±0.22% | **67.48±0.24%** | ✓ |
| DomainNet | CLIP ZS (G) | 94.72% | 93.49% | 95.94% | - |
| DomainNet | CLIP ZS (A) | 94.62% | 93.24% | 96.01% | - |
| DomainNet | CLIP LP | 94.39±0.04% | 95.03±0.07% | 93.75±0.02% | - |
| DomainNet | CLIP LP (ZS init) | 94.58±0.11% | 95.21±0.21% | 93.95±0.03% | - |
| DomainNet | WiSE-LP | 94.44±0.11% | 95.19±0.34% | 93.68±0.12% | - |
| DomainNet | VQGAN+CLIP | 94.67±0.09% | 95.54±0.09% | 93.83±0.10% | × |
| DomainNet | LADS | 95.27±0.14% | 95.33±0.33% | 95.21±0.09% | × |
| DomainNet | TEAM (P) (Ours) | 96.00±0.12% | 95.49±0.26% | 96.51±0.13% | ✓ |
| DomainNet | TEAM (G) (Ours) | **96.19±0.11%** | **95.59±0.27%** | **96.78±0.11%** | ✓ |

Table 1: In-domain (ID), out-of-domain (OOD) and the average (of ID and OOD) accuracy on **CUB-Paintings** and **DomainNet**. TEAM (P/G) denotes private or global modality alignment in Sec. 4.1 *Note that OOD is the major metrics, where the goal is to improve OOD performance without eroding ID accuracy.* TEAM outperforms baselines without training any augmentation network.

tween TEAM and baselines is larger. While the performance of TEAM-*invar.* is better or close to TEAM-*full*, the former requires significantly shorter training time during the second stage. Because TEAM-*invar.* benefits from extracting domain-invariant features and performing domain-invariant augmentations, where the number of total augmented embeddings is independent of the number of crafted unseen domain descriptions. Interestingly, we also observe that, even if without exact text descriptions of test domains (Tab. 2), by leveraging descriptions of randomly crafted unseen domains, we are able to achieve similar or even better performances compared to cases where exact test domain descriptions are given (Tab. 1). One of the potential factors that influence the performance is the similarity between crafted unseen domains and test domains. For instance, in experiments with CUB-Paintings, our crafted unseen domains are: "*sketch*", "*clipart*", "*product shot*", "*infographics*", "*Screenshots*", "*3D rendering*", "*cartoon*", and the test domain is "*photo*". There are crafted domains that share higher similarities with the test domain such as "*product shot*", which may contributes more to the model performance. Furthermore, this observation provides some insights for exploring cases where it is hard to exactly describe a test domain with language. The experiments imply that it may not be necessary to have the ground truth test domain description in advance. With randomly crafted domain descriptions, it is likely to have satisfying performances on test domains, especially when they are potentially relevant with test domains.

**Verbalizing unseen domains with different LLMs or existing templates**. In above experiments, we utilize ChatGPT to generate descriptions of crafted unseen domains. We also leverage other popular LLMs and existing prompt templates (Radford et al., 2021). Results in Appendix E show TEAM is robust across different sets of prompts.

**Extending to other pre-trained VL models**. We also use variants of CLIP as the backbone pre-trained VL model, where performances are consistent. Results are available in Appendix F.

## 5.5 ABLATIONS

We discuss how different choices of augmentation methods and extraction functions $g_{extract}(\cdot)$ influence the performance of TEAM. We propose two options for $g_{extract}(\cdot)$: meaning pooling and cosine autoencoder (Niu et al., 2022). Tab. 3 show the experiment results. While aligning modality direction is obviously better, the choices of $g_{extract}(\cdot)$ does not have a significant influence. However, performing domain-invariant augmentation with $g_{extract}(\cdot)$ notably reduces the training time.

## 5.6 CASE STUDY

We provide an intuitive visualization of the augmented embedding quality from LADS and ours. Because we are not able directly to transfer CLIP embeddings into a pixel-level image, similar

| Dataset | Method | Average | ID | OOD | Training-free (Stage-1) | Time (Stage-2) |
|---------|--------|---------|-----|-----|------------------------|----------------|
| CUB-Paintings | CLIP LP (ZS init) | 75.57±0.06% | 86.08±0.11% | 65.05±0.05% | - | - |
| CUB-Paintings | WiSE-LP | 73.27±0.22% | 81.74±0.34% | 64.80±0.10% | - | - |
| CUB-Paintings | LADS | 74.99±0.23% | 85.33±0.29% | 64.85±0.26% | × | 1 × |
| CUB-Paintings | TEAM–*full* (Ours) | 76.84±0.23% | 86.54±0.21% | 67.14±0.21% | ✓ | 1 × |
| CUB-Paintings | TEAM-*invar.* (Ours) | **77.16±0.19**% | **86.61±0.22**% | **67.71±0.23**% | ✓ | 0.23 × |
| DomainNet | CLIP LP (ZS init) | 94.58±0.11% | 95.21±0.21% | 93.95±0.03% | - | - |
| DomainNet | WiSE-LP | 94.44±0.11% | 95.19±0.34% | 93.68±0.12% | - | - |
| DomainNet | LADS | 94.97±0.25% | 95.29±0.33% | 94.65±0.09% | × | 1 × |
| DomainNet | TEAM-*full* (Ours) | 96.17±0.12% | **95.71±0.23**% | 96.55±0.18% | ✓ | 1 × |
| DomainNet | TEAM-*invar.* (Ours) | **96.18±0.14**% | 95.61±0.21% | **96.70±0.20**% | ✓ | 0.25 × |

Table 2: In-domain (ID), out-of-domain (OOD) and the average (of ID and OOD) accuracy on **CUB-Paintings** and **DomainNet**. *Note that OOD is the major metric, where the goal is to improve OOD accuracy without eroding ID accuracy.* We compare our models with most competing baselines in Tab. 1. We report results of our TEAM (G).

| Aug. Method | *Invar.* Mode | Average | ID | OOD | Training-free (Stage-1) | Time (Stage-2) |
|-------------|---------------|---------|-----|-----|------------------------|----------------|
| LADS | Mean Pooling | 74.99±0.23% | 85.33±0.29% | 64.85±0.26% | × | 1 × |
| Global Dir. | Mean Pooling | 74.67±0.22% | 85.21±0.21% | 64.12±0.21% | ✓ | 0.23 × |
| Modality Dir. | Mean Pooling | 77.16±0.19% | 86.61±0.22% | 67.71±0.23% | ✓ | 0.23 × |
| Modality Dir. | Cosine AutoEncoder | 77.18±0.21% | 86.62±0.18% | 67.74±0.23% | ✓ | 0.23 × |
| Modality Dir. | *None* | 76.84±0.23% | 86.54±0.21% | 67.14±0.21% | ✓ | 1 × |
| *Text only* | *None* | 75.98±0.23% | 85.90±0.21% | 66.06±0.21% | ✓ | 1 × |

Table 3: Performances of LADS and our variants on **CUB-Paintings** dataset. *Invar-Mode* refers to different methods to obtain domain-invariant representations. *None* means we do not use domain-invariant representations for augmentation. *Text only* means using text embeddings for training without being augmented to the image subspace. We report results of our TEAM (G).

to (Dunlap et al., 2023), we show some nearest neighboring results of augmented embeddings, which demonstrates our augmented embeddings are more informative. In Fig. 4, For each column, given a source image, the second and third row show the image from test dataset whose CLIP embedding is nearest to the augmented embedding (by LADS and TEAM) of the source image CLIP embedding.

While augmented embeddings by LADS retrieve embeddings of different classes by mistake, augmented embeddings by TEAM effectively preserve class information as well as domain information and successfully retrieve correct embeddings even when the visual difference is minor (last example in Fig. 4, where the task is to distinguish bird species).

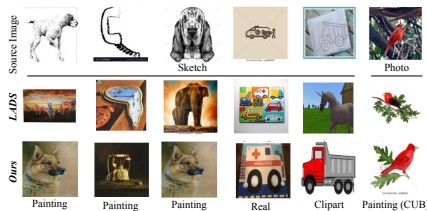

Figure 4: Nearest Neighboring Results.

# 6 LIMITATIONS AND FUTURE WORK

Because our text-driven augmentation requires text descriptions of target domains, it is challenging when some complicated natural domain shifts Koh et al. (2021) are hard to capture and express solely through language. As TEAM heavily relies on the multimodal embedding space of a pre-trained vision-language model (CLIP), it is also bottle-necked by the quality of that space. Even if we can verbalize above shifts with intricate and detailed descriptions, it can be hard for CLIP to accurately embed such complex semantics into its embedding space. Furthermore, the effectiveness of our data-free augmentation method is also influenced by the geometry of the embedding distribution.

We proposed a data-free method for text-driven embedding augmentation, and further combined it with our framework for the exploration of unseen domains. While our experiments reveal that it is possible to achieve satisfying performances on test domains without its exact text description, we are curious to know if crafting vague descriptions work for complicated domain shifts. Exploring how to verbalize intricate domains shifts is also an interesting topic. Besides, we also hope to see better use of multimodal embedding space in other ways.

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

## A    PROOF OF LEMMA 1

**Lemma 1** *Given an image-text pair $(\boldsymbol{x_i}, t_{source}; y_i)$, the text description $(t_{target}; y_i)$ from target domain, and text encoder $h^T(\cdot)$ and image encoder $h^I(\cdot)$ from a pre-trained vision language model , if the contrastive pre-training is moderately trained (e.g., the pre-trained CLIP (Radford et al., 2021)). we have:*

$$h^T(t_{\text{target}}; y_i) \cdot h^T(t_{\text{source}}; y_i) > h^T(t_{\text{target}}; y_i) \cdot h^I(\boldsymbol{x_i}) \tag{11}$$

$$\text{subject to } ||h^I(\boldsymbol{x_i})|| = ||h^T(t_{\text{source}}; y_i)|| = ||h^T(t_{\text{target}}; y_i)|| \tag{12}$$

*Note that Eq. (12) is naturally satisfied as the embeddings from CLIP (Radford et al., 2021) are normalized to a unit sphere.*

*Proof.* Recent work (Liang et al., 2022; Zhang et al., 2023) showed that, in the multimodal space of a vision language model pre-trained with a contrastive loss: embeddings are approximately clustered per modality. Furthermore, they (Liang et al., 2022; Zhang et al., 2023) reveal two geometric characteristics, which we formulate as two assumptions:

**Assumption 1.1** The modality gap between corresponding image and text embeddings can be approximately represented by a constant vector $\boldsymbol{g}$.

**Assumption 1.2** $\boldsymbol{g}$ is approximately orthogonal to the span of image embeddings and text embeddings.

Therefore, we have:

Let $h^I(\boldsymbol{x_i}) \approx \boldsymbol{g} + h^T(t_{\text{source}}; y_i)$ (from Assumption 1.1), combining Eq. (12) we obtain:

$$||h^I(\boldsymbol{x_i})||^2 = ||\boldsymbol{g} + h^T(t_{\text{source}}; y_i)||^2 = ||h^T(t_{\text{source}}; y_i)||^2$$

$$\Rightarrow ||h^T(t_{\text{source}}; y_i)||^2 = ||h^T(t_{\text{source}}; y_i)||^2 + ||\boldsymbol{g}||^2 + 2h^T(t_{\text{source}}; y_i) \cdot \boldsymbol{g}$$

$$\Rightarrow h^T(t_{\text{source}}; y_i) \cdot \boldsymbol{g} = -\frac{||\boldsymbol{g}||^2}{2} < 0$$

By Assumption 1.2, we have:

$$\boldsymbol{g} \cdot (h^T(t_{\text{target}}; y_i) - h^T(t_{\text{source}}; y_i)) \approx 0$$

Thus, $h^T(t_{\text{target}}; y_i) \cdot \boldsymbol{g} \approx h^T(t_{\text{source}}; y_i) \cdot \boldsymbol{g} < 0$.

As $\boldsymbol{g} \approx h^I(\boldsymbol{x_i}) - h^T(t_{\text{source}}; y_i)$ (from Lemma 1.1), combining it with the above equation by eliminating $\boldsymbol{g}$ , we have:

$$h^T(t_{\text{target}}; y_i) \cdot (h^I(\boldsymbol{x_i}) - h^T(t_{\text{source}}; y_i)) < 0,$$

$$\text{i.e., } h^T(t_{\text{target}}; y_i) \cdot h^T(t_{\text{source}}; y_i) > h^T(t_{\text{target}}; y_i) \cdot h^I(\boldsymbol{x_i})$$

Lemma 1 is proved.

## B  PROOF OF PROPOSITION 1

**Proposition 1**

$$\frac{\boldsymbol{z} - h^T(t_{\text{target}} ; y_i)}{||\boldsymbol{z} - h^T(t_{\text{target}} ; y_i)||} \cdot \frac{h^I(\boldsymbol{x_i}) - h^T(t_{\text{source}} ; y_i)}{||h^I(\boldsymbol{x_i}) - h^T(t_{\text{source}} ; y_i)||} = 1 \tag{13}$$

*Given an image-text pair $(\boldsymbol{x_i}, t_{source}; y_i)$, the corresponding text description from target domain $(t_{target}; y_i)$, and text and image encoders $h^T(\cdot)$ and $h^I(\cdot)$ from a pre-trained vision language model, Eq. (13) has a solution if the vision-language model is pre-trained with a contrastive loss when subject to:*

$$||\boldsymbol{z}|| = ||h^I(\boldsymbol{x_i})|| = ||h^T(t_{\text{source}}; y_i)|| = ||h^T(t_{\text{target}}; y_i)|| \tag{14}$$

*Note that Eq. (14) is naturally satisfied as the embeddings from CLIP (Radford et al., 2021) are normalized to a unit sphere.*

*Proof.* Consider Eq. (13), by observing the format of it, we have:

$$\boldsymbol{z} = \lambda(h^I(\boldsymbol{x_i}) - h^T(t_{\text{source}} ; y_i)) + h^T(t_{\text{target}} ; y_i), \tag{15}$$

where $\lambda$ is a non-negative coefficient. It is obvious because, combining Eq. (13) and Eq. (15), we have:

$$\frac{\boldsymbol{z} - h^I(\boldsymbol{x_i})}{||\boldsymbol{z} - h^I(\boldsymbol{x_i})||} \cdot \frac{h^I(\boldsymbol{x_i}) - h^T(t_{\text{source}} ; y_i)}{||h^I(\boldsymbol{x_i}) - h^T(t_{\text{source}} ; y_i)||}$$

$$= \frac{\lambda(h^I(\boldsymbol{x_i}) - h^T(t_{\text{source}} ; y_i))}{||\lambda(h^I(\boldsymbol{x_i}) - h^T(t_{\text{source}} ; y_i))||} \cdot \frac{h^I(\boldsymbol{x_i}) - h^T(t_{\text{source}} ; y_i)}{||h^I(\boldsymbol{x_i}) - h^T(t_{\text{source}} ; y_i)||}$$

$$= \frac{\lambda||(h^I(\boldsymbol{x_i}) - h^T(t_{\text{source}} ; y_i))||^2}{||\lambda(h^I(\boldsymbol{x_i}) - h^T(t_{\text{source}} ; y_i))||^2}$$

$$= \frac{\lambda||(h^T(t_{\text{target}} ; y_i) - h^T(t_{\text{source}} ; y_i))||^2}{||\lambda|| \cdot ||(h^T(t_{\text{target}} ; y_i) - h^T(t_{\text{source}} ; y_i))||^2}$$

$$= \frac{\lambda}{||\lambda||},$$

$$= 1 \text{ if } \lambda > 0$$

Then we solve $\lambda$ as follows. Combining Eq. (15) with the constraint Eq. (14), we have:

$$||h^T(t_{\text{target}}; y_i)|| = ||\lambda(h^I(\boldsymbol{x_i}) - h^T(t_{\text{source}} ; y_i)) + h^T(t_{\text{target}} ; y_i)||$$

$$\Rightarrow ||h^T(t_{\text{target}}; y_i)||^2 = ||\lambda(h^I(\boldsymbol{x_i}) - h^T(t_{\text{source}} ; y_i)) + h^T(t_{\text{target}} ; y_i)||^2$$

$$\Rightarrow (h^T(t_{\text{target}}; y_i))^2 = (\lambda(h^I(\boldsymbol{x_i}) - h^T(t_{\text{source}} ; y_i)) + h^T(t_{\text{target}} ; y_i))^2,$$

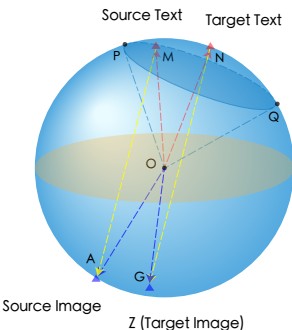 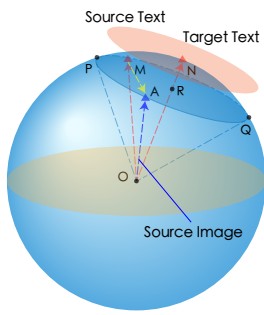

Figure 5: Visualization of embeddings in feature space. The red plane (right) is the tangent plane passing through point $N$. **Left** (success case): Eq. (20) is satisfied, i.e., the target text embedding is closer to the source text embedding than the source image embedding, where Eq. (21) has a solution. **Right** (failure/boundary case): $\overrightarrow{ON} \cdot \overrightarrow{OM} = \overrightarrow{ON} \cdot \overrightarrow{OA}$, where Eq. (20) is not satisfied and Eq. (21) does not have a solution. Recent work (Liang et al., 2022; Zhang et al., 2023) validated that the distribution of CLIP (Radford et al., 2021) embeddings is consistent with the left figure.

$$\text{subject to } \lambda > 0$$

Organizing the above equation, we have:

$$((h^I(\boldsymbol{x_i}) - h^T(t_{\text{source}}\,;y_i)) + h^T(t_{\text{target}}\,;y_i))^2\lambda^2 + 2(\lambda(h^I(\boldsymbol{x_i}) - h^T(t_{\text{source}}\,;y_i)) +$$
$$h^T(t_{\text{target}}\,;y_i)) \cdot h^T(t_{\text{target}}\,;y_i)\lambda = 0 \tag{16}$$

The above equation has two solutions:

$$\begin{cases} \lambda_0 = 0 \text{ (excluded)} \\ \lambda_1 = \dfrac{-2h^T(t_{\text{target}}\,;y_i) \cdot (h^I(\boldsymbol{x_i}) - h^T(t_{\text{source}}\,;y_i))}{(h^I(\boldsymbol{x_i}) - h^T(t_{\text{source}}\,;y_i))^2} \end{cases}, \tag{17}$$

where $\lambda_0$ is excluded. Note that $\lambda_1$ must be non-negative. However, it is not guaranteed with arbitrary encoders $h^T(\cdot)$ and $h^I(\cdot)$.

Consider Eq. (17):

$$\lambda_1 > 0 \iff -2h^T(t_{\text{target}}\,;y_i) \cdot (h^I(\boldsymbol{x_i}) - h^T(t_{\text{source}}\,;y_i)) > 0$$
$$\iff h^T(t_{\text{target}}\,;y_i) \cdot h^T(t_{\text{source}};y_i) - h^T(t_{\text{target}}\,;y_i) \cdot h^I(\boldsymbol{x_i}) > 0$$

By Lemma 1, one can derive that the inequality above holds when $h^T(\cdot)$ and $h^I(\cdot)$ come from a vision-language model with contrastive pre-training (Eq. (11)).

Therefore, Eq. (13) must have an analytical solution $\boldsymbol{z}$ under the constraint Eq. (14) as long as the vision-language model is moderately trained. we complete the proof.

Combining Eq. (15) and Eq. (17), we finally have:

$$\boldsymbol{z} = \frac{-2h^T(t_{\text{target}}\,;y_i) \cdot (h^I(\boldsymbol{x_i}) - h^T(t_{\text{source}}\,;y_i))}{(h^I(\boldsymbol{x_i}) - h^T(t_{\text{source}}\,;y_i))^2} \cdot (h^I(\boldsymbol{x_i}) - h^T(t_{\text{source}}\,;y_i)) + h^T(t_{\text{target}}\,;y_i) \tag{18}$$

### B.1 VISUALIZATION EXPLANATION OF PROPOSITION 1

We also provide the explanation and proof of the existence of solution to Eq. (13) from a geometrical perspective. In Fig. 5, red and blue arrows are text and image embeddings, respectively. In Fig. 5 right, we draw a circular section $\odot R$ of the unit sphere, where $\overrightarrow{ON}$ passes through the center $R$, and $M$ lies on the circumference. It can be observed that only when $A$ is located on the spherical cap below $\odot R$ (Figure 5 left), we can find $G$ on the sphere such that $\overrightarrow{NG}$ and $\overrightarrow{MA}$ are parallel, i.e., Eq. (13) has a solution. This is because, considering the boundary case (Fig. 5 right), when point $A$ lies on the circumference of $\odot R$, we have: $\overrightarrow{MA}$ is parallel to the tangent plane passing through

point $N$. In this case, we happen to be not able to find a point $G$ on the spherical surface such that $\overrightarrow{NG} \parallel \overrightarrow{MA}$. If point $A$ is further moved towards to the top of the upper spherical cap, it is obvious that such $G$ does not exit. Only when $A$ is located on the lower spherical cap (Fig. 5 left), we are able to find a point $G$ such that $\overrightarrow{NG} \parallel \overrightarrow{MA}$. So far we know Eq. (13) has a solution when $A$ is located on the lower spherical cap. Next, we show it is guaranteed by Eq. (11).

If Eq. (11) holds, we have:

$$h^T(t_{\text{target}}; y_i) \cdot h^T(t_{\text{source}}; y_i) > h^T(t_{\text{target}}; y_i) \cdot h^I(\boldsymbol{x_i})$$

$$\Rightarrow \frac{h^T(t_{\text{target}}; y_i) \cdot h^T(t_{\text{source}}; y_i)}{||1|| \cdot ||1||} > \frac{h^T(t_{\text{target}}; y_i) \cdot h^I(\boldsymbol{x_i})}{||1|| \cdot ||1||}$$

$$\Rightarrow \frac{h^T(t_{\text{target}}; y_i) \cdot h^T(t_{\text{source}}; y_i)}{||h^T(t_{\text{target}}; y_i)|| \cdot ||h^T(t_{\text{source}}; y_i)||} > \frac{h^T(t_{\text{target}}; y_i) \cdot h^I(\boldsymbol{x_i})}{||h^T(t_{\text{target}}; y_i)|| \cdot ||h^I(\boldsymbol{x_i})||}$$

Thus Lemma 1 indicates the angle between the target text embedding and the source text embedding is smaller than that between the target text embedding and the source image embedding. It is obvious from Fig. 5 (right) that, if Lemma 1 holds, the point $A$ must be located on the lower spherical cap.

## C    PROOF OF PROPOSITION 2

The image-to-text loss function of CLIP is:

$$\mathcal{L}_{I2T} = -\frac{1}{N} \sum_{i=1}^{N} \log \frac{\exp\left(h^I(\boldsymbol{x_i}) \cdot h^T(t_{\text{training}}; y_{\mathbf{i}})/\tau\right)}{\sum_{j=1}^{N} \exp\left(h^I(\boldsymbol{x_i}) \cdot h^T(t_{\text{training}}; y_{\mathbf{j}})/\tau\right)}, \tag{19}$$

Where $N$ is the number of training samples and $\tau$ is the temperature scalar.

**Lemma 2** *Given an image-text pair $(\boldsymbol{x_i}, t_{training}; y_i)$, the text description $(t_{unseen}; y_i)$ from target domain, and text encoder $h^T(\cdot)$ and image encoder $h^I(\cdot)$ from a pre-trained vision language model (CLIP (Radford et al., 2021), if the contrastive pre-training is perfect. we have:*

$$h^I(\boldsymbol{x_i}) \cdot h^T(t_{\text{training}}; y_i) > h^I(\boldsymbol{x_i}) \cdot h^T(t_{\text{unseen}}; y_i) \tag{20}$$

*Proof.* As the contrastive pre-training is perfect, we have $\mathcal{L}_{I2T} \to 0$.

For each $i$, let $\log \frac{\exp\left(h^I(\boldsymbol{x_i}) \cdot h^T(t_{\text{training}}; y_{\mathbf{i}})/\tau\right)}{\sum_{j=1}^{N} \exp\left(h^I(\boldsymbol{x_i}) \cdot h^T(t_{\text{training}}; y_{\mathbf{j}})/\tau\right)} \to 0$, we have:

$$\frac{\exp\left(h^I(\boldsymbol{x_i}) \cdot h^T(t_{\text{training}}; y_{\mathbf{i}})/\tau\right)}{\sum_{j=1}^{N} \exp\left(h^I(\boldsymbol{x_i}) \cdot h^T(t_{\text{training}}; y_{\mathbf{j}})/\tau\right)} = \alpha, \text{where } \alpha \to 1.$$

$$\Rightarrow \exp\left(h^I(\boldsymbol{x_i}) \cdot h^T(t_{\text{training}}; y_{\mathbf{i}})/\tau\right) \cdot (1 - \alpha) = \sum_{j=1}^{N} \exp\left(h^I(\boldsymbol{x_i}) \cdot h^T(t_{\text{training}}; y_{\mathbf{j}})/\tau\right) \cdot \alpha,$$

where $\alpha \to 1$. Then we have, for each $j \neq i$, $\exp\left(h^I(\boldsymbol{x_i}) \cdot h^T(t_{\text{training}}; y_{\mathbf{j}})/\tau\right) \to 0$, which indicates the cosine similarity of the negative image-text pair is less than that of the positive pair, i.e., Eq. (20).

**Proposition 2**

$$\frac{\boldsymbol{z} - h^I(\boldsymbol{x_i})}{||\boldsymbol{z} - h^I(\boldsymbol{x_i})||} \cdot \frac{h^T(t_{\text{unseen}}^k; y_i) - h^T(t_{\text{training}}; y_i)}{||h^T(t_{\text{unseen}}^k; y_i) - h^T(t_{\text{training}}; y_i)||} = 1 \tag{21}$$

*Given an image-text pair $(\boldsymbol{x_i}, t_{training}; y_i)$, the corresponding text description from unseen domain $(t_{unseen}; y_i)$, and text and image encoders $h^T(\cdot)$ and $h^I(\cdot)$ from a pre-trained vision language model, Eq. (21) has a solution if the contrastive learning of the vision-language model is perfect when subject to:*

$$||\boldsymbol{z}|| = ||h^I(\boldsymbol{x_i})|| = ||h^T(t_{\text{training}}; y_i)|| = ||h^T(t_{\text{unseen}}; y_i)|| \tag{22}$$

*Note that Eq. (22) is naturally satisfied as the embeddings from CLIP (Radford et al., 2021) are normalized to a unit sphere.*

*Proof.* Consider Eq. 21, by observing the format of it, we have:

$$z = \lambda(h^T(t^k_{\text{unseen}}\,;y_i) - h^T(t_{\text{training}}\,;y_i)) + h^I(\boldsymbol{x_i}),\tag{23}$$

where $\lambda$ is a non-negative coefficient. It is obvious because, combining Eq. (21) and Eq. (23), we have:

$$\frac{z - h^I(\boldsymbol{x_i})}{\|z - h^I(\boldsymbol{x_i})\|} \cdot \frac{h^T(t^k_{\text{unseen}}\,;y_i) - h^T(t_{\text{training}}\,;y_i)}{\|h^T(t^k_{\text{unseen}}\,;y_i) - h^T(t_{\text{training}}\,;y_i)\|}$$

$$= \frac{\lambda(h^T(t^k_{\text{unseen}}\,;y_i) - h^T(t_{\text{training}}\,;y_i))}{\|\lambda(h^T(t^k_{\text{unseen}}\,;y_i) - h^T(t_{\text{training}}\,;y_i))\|} \cdot \frac{h^T(t^k_{\text{unseen}}\,;y_i) - h^T(t_{\text{training}}\,;y_i)}{\|h^T(t^k_{\text{unseen}}\,;y_i) - h^T(t_{\text{training}}\,;y_i)\|}$$

$$= \frac{\lambda\|(h^T(t^k_{\text{unseen}}\,;y_i) - h^T(t_{\text{training}}\,;y_i))\|^2}{\|\lambda(h^T(t^k_{\text{unseen}}\,;y_i) - h^T(t_{\text{training}}\,;y_i))\|^2}$$

$$= \frac{\lambda\|(h^T(t^k_{\text{unseen}}\,;y_i) - h^T(t_{\text{training}}\,;y_i))\|^2}{\|\lambda\| \cdot \|(h^T(t^k_{\text{unseen}}\,;y_i) - h^T(t_{\text{training}}\,;y_i))\|^2}$$

$$= \frac{\lambda}{\|\lambda\|},$$

$$= 1 \text{ if } \lambda > 0$$

Then we solve $\lambda$ as follows. Combining Eq. (23) with the constraint Eq. (22), we have:

$$\|h^T(t_{\text{training}}; y_i)\| = \|\lambda(h^T(t^k_{\text{unseen}}\,;y_i) - h^T(t_{\text{training}}\,;y_i)) + h^I(\boldsymbol{x_i})\|$$

$$\Rightarrow \|h^T(t_{\text{training}}; y_i)\|^2 = \|\lambda(h^T(t^k_{\text{unseen}}\,;y_i) - h^T(t_{\text{training}}\,;y_i)) + h^I(\boldsymbol{x_i})\|^2$$

$$\Rightarrow (h^T(t_{\text{training}}; y_i))^2 = (\lambda(h^T(t^k_{\text{unseen}}\,;y_i) - h^T(t_{\text{training}}\,;y_i)) + h^I(\boldsymbol{x_i}))^2,$$

$$\text{subject to } \lambda > 0$$

Organizing the above equation, we have:

$$((h^T(t^k_{\text{unseen}}\,;y_i) - h^T(t_{\text{training}}\,;y_i)) + h^I(\boldsymbol{x_i}))^2\lambda^2 + 2(\lambda(h^T(t^k_{\text{unseen}}\,;y_i) - h^T(t_{\text{training}}\,;y_i)) +$$

$$h^I(\boldsymbol{x_i})) \cdot h^I(\boldsymbol{x_i})\lambda = 0$$

$$\tag{24}$$

Above equation has two solutions:

$$\begin{cases} \lambda_0 = 0 \text{ (excluded)} \\ \lambda_1 = \dfrac{-2h^I(\boldsymbol{x_i}) \cdot (h^T(t^k_{\text{unseen}}\,;y_i) - h^T(t_{\text{training}}\,;y_i))}{(h^T(t^k_{\text{unseen}}\,;y_i) - h^T(t_{\text{training}}\,;y_i))^2} \end{cases},\tag{25}$$

where $\lambda_0$ is excluded. Note that $\lambda_1$ must be non-negative. However, it is not guaranteed with arbitrary encoders $h^T(\cdot)$ and $h^I(\cdot)$.

Consider Eq. (25):

$$\lambda_1 > 0 \iff -2h^I(\boldsymbol{x_i}) \cdot (h^T(t^k_{\text{unseen}}\,;y_i) - h^T(t_{\text{training}}\,;y_i)) > 0$$

$$\iff h^I(\boldsymbol{x_i}) \cdot h^T(t_{\text{training}}; y_i) - h^I(\boldsymbol{x_i}) \cdot h^T(t_{\text{unseen}}; y_i) > 0$$

By Lemma 2, one can derive that the right equation above holds when $h^T(\cdot)$ and $h^I(\cdot)$ come from a vision-language model with perfect contrastive pre-training (Eq. (20)). Thus Eq. (21) must have an analytical solution $z$ under the constraint Eq. (22) as long as the vision-language model is perfect. we complete the proof.

Combining Eq. (23) and Eq. (25), we finally have:

$$z = \frac{-2h^I(\boldsymbol{x_i}) \cdot (h^T(t^k_{\text{unseen}}\,;y_i) - h^T(t_{\text{training}}\,;y_i))}{(h^T(t^k_{\text{unseen}}\,;y_i) - h^T(t_{\text{training}}\,;y_i))^2} \cdot (h^T(t^k_{\text{unseen}}\,;y_i) - h^T(t_{\text{training}}\,;y_i)) + h^I(\boldsymbol{x_i})$$

$$\tag{26}$$

# D EXPERIMENT

## D.1 BASELINES

*CLIP ZS* is proposed by Radford et al. (2021) for zero-shot classification. While *CLIP ZS generic (G)* only uses class name as the text prompt (e.g. "camera"), *CLIP ZS adaptive (A)* customizes text prompts for specific domains (e.g. "a painting of an airplane").

*CLIP LP* applies a linear classifier to the CLIP image embeddings. *CLIP LP (ZS init)* initializes the linear classifier with the text embeddings.

*WiSE-LP* (Wortsman et al., 2022) is an ensembling method which fine-tunes a CLIP model and does a weighted average of the fine-tuned model's weights with the original. Following LADS, we avoid fine-tuning the entire backbone and instead ensembled the classifier with the linear classifier probe as introduced by Wortsman et al. (2022).

*VQGAN + CLIP* (Crowson et al., 2022) augments raw images in pixel space with a VQGAN (Esser et al., 2021) trained with CLIP. With a text prompt and an image, a style transfer (augmentation) target domain is performed to the source image. Then a linear probe is fitted on both source images and augmented images. According to LADS, due to the amount of time and compute required to generate images, the baseline only runs DomainNet and approximately 15% of the training dataset is augmented.

*LADS* (Dunlap et al., 2023) is an augmentation-based method. Different from (Crowson et al., 2022), LADS leverages a pre-trained vision-language model to obtain text and image embeddings and performs the latent augmentation in the embedding space. Then a linear classifier is fitted on both original embeddings and augmented embeddings.

Note that we do not consider (Cho et al., 2023) as a baseline for multiple reasons. First, we work on different settings. (Cho et al., 2023) works on source-free domain generalization, where *no* source images are available during training. In this setting, the model particularly relies on the learnt general knowledge of common classes (e.g., dog, car, etc.) in CLIP, which however, cannot learn visual features of intricate classes from images.

In contrast, we extend the problem setting of LADS (Dunlap et al., 2023), where source images are available and the goal is to learn from source images as well as augmented target images for domain generalization. We are working on different problems and are orthogonal to each other. *Source-free* in Tab. 4 can be regarded as an approximation of the upper-bound of (Cho et al., 2023) in our setting. This is because, while (Cho et al., 2023) are substantially learning the learnable embeddings of different domain descriptions, we directly give the ground truth text descriptions for *Source-free* in the following table. It is significantly worse than other methods because, without source images, it mainly relies on the learnt general knowledge of common classes (e.g., dog, car, etc.) in CLIP for generalization. However, classes in CUB-Paintings dataset consist of fine-grained bird names, which pre-trained CLIP cannot recognize.

| Aug. Method | *Invar.* Mode | Average | ID | OOD | Training-free (Stage-1) | Time (Stage-2) |
|---|---|---|---|---|---|---|
| LADS | Mean Pooling | 74.99±0.23% | 85.33±0.29% | 64.85±0.26% | × | 1 × |
| *Source-free* | *None* | 50.07±0.18% | 49.57±0.19% | 50.57±0.19% | – | – |
| Modality Dir. (Ours) | Mean Pooling | 77.16±0.19% | 86.61±0.22% | 67.71±0.23% | ✓ | 0.23 × |
| Modality Dir. (Ours) | Cosine AutoEncoder | 77.18±0.21% | 86.62±0.18% | 67.74±0.23% | ✓ | 0.23 × |
| Modality Dir. (Ours) | *None* | 76.84±0.23% | 86.54±0.21% | 67.14±0.21% | ✓ | 1 × |
| *Text only* (Ours) | *None* | 75.98±0.23% | 85.90±0.21% | 66.06±0.21% | ✓ | 1 × |

Table 4: Performances of LADS and our variants and *Source-free* on **CUB-Paintings** dataset.

## D.2 DETAILED EXPERIMENTAL SETTINGS

**DomainNet.** Following (Dunlap et al., 2023), we use a specific split (Tan et al., 2020) of the DomainNet (Peng et al., 2019) dataset which contains 40 most common classes from 4 domains: 'sketch', 'real', 'clipart', and 'painting'. Following (Tan et al., 2020; Kumar et al., 2022; Dunlap et al., 2023) , we train on sketches and evaluate on the three other domains.

In the LADS setting, i.e., we have exact test domain descriptions during training, we follow the LADS and use the test domain descriptions in Tab. 5 for augmentation.

| Prompts |
| --- |
| `a clipart of a {...}` |
| `a painting of a {...}` |
| `a realistic photo of a {...}` |

Table 5: Test domain descriptions

In our text-driven domain generalization scenario, where we do not have text domain descriptions, we randomly prompted unseen domain descriptions by asking ChatGPT to give us several domain names and manually select from them. Note that test domain names are excluded. In particular, we use domain descriptions in Tab. 6 for augmentation.

| Prompts |
| --- |
| `an image of a {...}` |
| `a product shot of a {...}` |
| `an infographics of a {...}` |
| `a screenshot of a {...}` |
| `a 3D rendering of a {...}` |
| `a cartoon of a {...}` |

Table 6: Crafted domain descriptions

Tab. 7 provides an example of the question we ask to obtain the answers:

| |
| --- |
| **User**: "*List some common domains of an image, such as sketches.*" |
| **Large Language Model**: "*icon, 3D rendering, clipart, painting, image, cartoon, ...*" |

Table 7: Obtaining crafted domains with a language model

**CUB-Paintings.** It combines CUB-200 (Wah et al., 2011) and CUB-200-Paintings (Wang et al., 2020), where there are 200 different bird species from "photo" and "painting". Following (Dunlap et al., 2023), we train on phtots and evaluate on painting.

In the LADS setting, i.e., we have exact test domain descriptions during training, we follow the LADS and use the test domain descriptions in Tab. 8 for augmentation.

In our text-driven domain generalization scenario, where we do not have text domain descriptions, we randomly prompted unseen domain descriptions by asking ChatGPT to give us several domain names and manually select from them. Note that test domain names are excluded. In particular, we use prompts in Tab. 9 for augmentation.

### D.3 HYPERPARAMETERS

For the LADS baseline, we adopt the default hyperparameters used in the original paper. In general, otherwise specified, we set the learning rate to 0.001, the weight decay to 0.05, and run experiments for 50 epochs. For DomainNet, we set the learning rate to 0.0001. We set the weight $\alpha$ to 0.5.

## E PERFORMANCE UNDER DIFFERENT CRAFTED DOMAIN DESCRIPTIONS

### E.1 IMAGENET PROMPT TEMPLATES

We also investigate if our method is robust towards difference choices of different groups of domain descriptions.

| Prompts |
|---|
| a realistic photo of a {...} |

Table 8: Test domain description

| Prompts |
|---|
| a sketch of a {...} |
| a clipart of a {...} |
| a product shot of a {...} |
| a infographics of a {...} |
| a screenshot of a {...} |
| a 3D rendering of a {...} |
| a cartoon of a {...} |

Table 9: Crafted domain description

To this end, besides selected domain descriptions given by ChatGPT (i.e., crafted templates), we also experiment with existing popular prompt templates. ImageNet Prompt Templates are proposed in CLIP (Radford et al., 2021), where there are 80 different context prompts. The templates are shown in Tab. 12.

It is practical in real-world applications to use such existing popular open templates. In practice, we can use a great number of possible unseen domains, which may cover the test domains or at least share larger similarities with test domains so that we can achieve satisfying performance on test domains. One potential risk is that, while desired domains may be included, noises can be involved as well, and chances are that the number of totally unrelated domains is notably larger than related ones. Therefore, we are curious to know how TEAM performs with a large number of prompts.

Results are available in Tab. 10, where we can see using crafted templates or ImageNet templates does not make a substantial difference. It manifests our method is robust against different sets of domain descriptions. In real-world applications, we can simply use existing templates such as ImageNet Templates for convenience.

### E.2 PROMPTS FROM DIFFERENCE LARGE LANGUAGE MODELS

In previous examples (Tab. 7), we ask ChatGPT(GPT3.5) to provide prompts during training. We also asked other language models to provide unseen prompts for augmentation during training, including GPT4 (OpenAI, 2023), New Bing[1], Bard[2]. Results are shown in Tab. 11. Consistent with the observations in Tab. 10, different sets of prompts do not make a significant different on the model performance.

## F PERFORMANCE UNDER DIFFERENT BACKBONES

Following LADS (Dunlap et al., 2023), we use CLIP (Radford et al., 2021) as our backbone model. We also explore the potential to replace it with other pre-trained vision language models. For simplicity, we experiment with FILIP (Yao et al., 2022), a variant of CLIP that shares the similar structure. Results are available in Tab. 11. FILIP demonstrates comparable or slightly better performance compared with CLIP.

---

[1]https://www.bing.com/new
[2]https://bard.google.com/

| Dataset | Method | Average | ID | OOD | Prompt Templates |
|---|---|---|---|---|---|
| CUB-Paintings | CLIP LP (ZS init) | 75.57±0.06% | 86.08±0.11% | 65.05±0.05% | - |
| CUB-Paintings | WiSE-LP | 73.27±0.22% | 81.74±0.34% | 64.80±0.10% | - |
| CUB-Paintings | LADS | 74.99±0.23% | 85.33±0.29% | 64.85±0.26% | Crafted Templates |
| CUB-Paintings | TEAM-*invar.* (Ours) | 77.16±0.19% | 86.61±0.22% | **67.71±0.23**% | Crafted Templates |
| CUB-Paintings | TEAM-*invar.* (Ours) | **77.23±0.23**% | **86.74±0.20**% | **67.71±0.22**% | ImageNet Templates |
| DomainNet | CLIP LP (ZS init) | 94.58±0.11% | 95.21±0.21% | 93.95±0.03% | - |
| DomainNet | WiSE-LP | 94.44±0.11% | 95.19±0.34% | 93.68±0.12% | - |
| DomainNet | LADS | 94.97±0.25% | 95.29±0.33% | 94.65±0.09% | Crafted Templates |
| DomainNet | TEAM-*invar.* (Ours) | **96.18 ±0.14**% | **95.61±0.21**% | 96.70±0.20% | Crafted Templates |
| DomainNet | TEAM-*invar.* (Ours) | 96.15 ±0.13% | 95.51±0.24% | **96.81±0.11**% | ImageNet Templates |

Table 10: In-domain (ID), out-of-domain (OOD) and the average (of ID and OOD) accuracy on **CUB-Paintings** and **DomainNet** with different prompt templates. Note that we do not use ImageNet templates with LADS because LADS requires training an augmentation network for each domain, which takes too much time as there are 80 prompts in total.

| Method | Average | ID | OOD | Source of Prompts | Backbone |
|---|---|---|---|---|---|
| TEAM-*invar.* (Ours) | 77.16±0.19% | 86.61±0.22% | 67.71±0.23% | ChatGPT (GPT3.5) | CLIP |
| TEAM-*invar.* (Ours) | 77.13±0.20% | 86.87±0.19% | 67.38±0.23% | GPT4 | CLIP |
| TEAM-*invar.* (Ours) | 77.09±0.22% | 87.03±0.21% | 67.16±0.22% | New Being | CLIP |
| TEAM-*invar.* (Ours) | 77.11±0.23% | 86.85±0.20% | 67.36±0.22% | Bard | CLIP |
| TEAM-*invar.* (Ours) | 77.23±0.21% | 86.74±0.20% | 67.71±0.19% | ImageNet Templates | CLIP |
| TEAM-*invar.* (Ours) | 77.23±0.21% | 86.67±0.22% | 67.78±0.20% | ChatGPT (GPT3.5) | FILIP |
| TEAM-*invar.* (Ours) | 77.18±0.19% | 86.92±0.25% | 67.43±0.24% | GPT4 | FILIP |
| TEAM-*invar.* (Ours) | 77.15±0.24% | 87.08±0.19% | 67.22±0.19% | New Being | FILIP |
| TEAM-*invar.* (Ours) | 77.17±0.22% | 86.9±0.24% | 67.43±0.20% | Bard | FILIP |
| TEAM-*invar.* (Ours) | 77.28±0.20% | 86.8±0.23% | 67.77±0.21% | ImageNet Templates | FILIP |

Table 11: In-domain (ID), out-of-domain (OOD) and the average (of ID and OOD) accuracy on **CUB-Paintings** with different sources of crafted prompts templates and backbones.

| Prompts |
|---|
| a bad photo of a {...} |
| a photo of many {...} |
| a sculpture of a {...} |
| a photo of the hard to see {...} |
| a low resolution photo of the {...} |
| a rendering of a {...} |
| graffiti of a {...} |
| a bad photo of the {...} |
| a cropped photo of the {...} |
| a tattoo of a {...} |
| the embroidered {...} |
| a photo of a hard to see {...} |
| a bright photo of a {...} |
| a photo of a clean {...} |
| a photo of a dirty {...} |
| a dark photo of the {...} |
| a drawing of a {...} |
| a photo of my {...} |
| the plastic {...} |
| a photo of the cool {...} |
| a close-up photo of a {...} |

| |
|---|
| a black and white photo of the {...} |
| a painting of the {...} |
| a painting of a {...} |
| a pixelated photo of the {...} |
| a sculpture of the {...} |
| a bright photo of the {...} |
| a cropped photo of a {...} |
| a plastic {...} |
| a photo of the dirty {...} |
| a jpeg corrupted photo of a {...} |
| a blurry photo of the {...} |
| a photo of the {...} |
| a good photo of the {...} |
| a rendering of the {...} |
| a  in a video game {...} |
| a photo of one {...} |
| a doodle of a {...} |
| a close-up photo of the {...} |
| a photo of a {...} |
| the origami {...} |
| the  in a video game {...} |
| a sketch of a {...} |
| a doodle of the {...} |
| a origami {...} |
| a low resolution photo of a {...} |
| the toy {...} |
| a rendition of the {...} |
| a photo of the clean {...} |
| a photo of a large {...} |
| a rendition of a {...} |
| a photo of a nice {...} |
| a photo of a weird {...} |
| a blurry photo of a {...} |
| a cartoon {...} |
| art of a {...} |
| a sketch of the {...} |
| a embroidered {...} |
| a pixelated photo of a {...} |
| itap of the {...} |
| a jpeg corrupted photo of the {...} |
| a good photo of a {...} |
| a plushie {...} |
| a photo of the nice {...} |
| a photo of the small {...} |
| a photo of the weird {...} |
| the cartoon {...} |
| art of the {...} |
| a drawing of the {...} |
| a photo of the large {...} |
| a black and white photo of a {...} |
| the plushie {...} |
| a dark photo of a {...} |
| itap of a {...} |
| graffiti of the {...} |
| a toy {...} |
| itap of my {...} |

| |
|---|
| a photo of a cool {...} |
| a photo of a small {...} |
| a tattoo of the {...} |

Table 12: ImageNet Prompt Templates

