# OpenReview forum: "Extending to New Domains without Visual and Textual Oracles"
_ICLR.cc/2024/Conference — Submitted to ICLR 2024_

### Official Review · Reviewer_GcYV · 2023-10-27

**Soundness:** 2 fair
**Presentation:** 3 good
**Contribution:** 2 fair
**Rating:** 3
**Confidence:** 4

**Summary:**

In this paper, the authors introduce an approach aimed at generalizing from a single domain, represented by natural object images, to previously unseen domains, including objects presented as paintings and sketches. Their method involves shifting visual embeddings using relative differences of textual embeddings from the original domain (a picture of a cat) to potential unseen test domains (a painting of a cat), thereby enabling feature-space augmentations.

The authors conduct evaluations of their proposal on datasets such as CUB and DomainNet. The results show minor improvements over the competitive LADS baseline. This work addresses an important challenge in domain adaptation by expanding the model's capabilities to novel domains, demonstrating its potential in real-world scenarios involving diverse visual data representations.


------------------------- After Rebuttal ----------------------------------------------------------------------------------------------

I thank the authors for their kind, lengthy, well written rebuttal. I carefully went through all the other reviews and post rebuttal text. However, it fails to upgrade my score upwards for three reasons:

- The authors acknowledge the similarity to CGAP, with the apparent difference being training-based vs. training-free. This opens up a whole new set of baselines to compare against, for training-free CLIP.
- However, I still see no update or discussion of this important paper.
- Lack of significant improvement over LADS is still a concern to consider.

**Strengths:**

S1:
The paper is generally well-written and easy to follow, although there are instances where it may overly emphasize LADS rather than the paper's ultimate goal. Despite this, it remains a good read overall, providing a clear presentation of the research.

S2:
The paper takes on a highly challenging task: single-domain generalization. In the context of this demanding objective, CLIP has established itself as a robust tool, and this paper contributes to this ongoing trend.

**Weaknesses:**

W1 Method:

A significant concern arises regarding the similarity between this paper and CLIP the GAP (CGAP) [1], which the authors fail to acknowledge or cite. Both papers share fundamental observations: that various domains of the same object differ by constant vectors. Additionally, they both tackle the challenge of single-domain generalization. The methodological resemblance is evident: CGAP performs semantic augmentations by computing relative differences in original and target test domains of text embeddings, using these as pseudo targets to generate visual features of the target domains. The evaluation contexts differ slightly: CGAP explores weather attributes, while this paper focuses on DomainNet/CUB. Furthermore, both papers demonstrate that precise domain names are unnecessary for strong performance on unseen domains, as illustrated by CGAP's "random" augmentations in Table 7.

In light of these similarities, it appears that this paper functions as an application on CUB and DomainNet rather than a substantially novel contribution.

[1] Vidit, Vidit, Martin Engilberge, and Mathieu Salzmann. "CLIP the Gap: A Single Domain Generalization Approach for Object Detection." CVPR, 2023. (https://openaccess.thecvf.com/content/CVPR2023/papers/Vidit_CLIP_the_Gap_A_Single_Domain_Generalization_Approach_for_Object_CVPR_2023_paper.pdf)

W2 Result:

A noteworthy concern arises from the marginal improvement over the most competitive baseline, LADS, as evidenced by Table 1 (0.78% improvement on CUB, 0.92% improvement on DomainNet). This minor improvement raises doubts about the merit and significance of the proposed method, especially considering the striking similarities to existing work, such as CGAP. The paper's originality and contribution require further clarification and validation to convincingly demonstrate its value in the domain of single-domain generalization.

Typos:

- [LADS reference]: "Using language to *extend* to unseen domains," International Conference on Learning Representations (ICLR), 2023.

- "Benefiting from the learned multimodal embedding space in pre-trained large-scale vision-language (VL) models (Radford et al., 2021; Jia et al., 2021), *we* achieve..."

**Questions:**

N/A

---

> ### Author Response · Authors · 2023-11-16
> **Response to Reviewer GcYV (Part 1)**
>
> We sincerely thank you for your review and helpful comments!
> ## Q1 The similarity between this paper and CLIP the GAP (CGAP) [1].
> Thanks for pointing out this related work. We will include and discuss it in our final paper. Aftering carefully checking it, we clarity that, although we may share similarity in working under domain generalization with CGAP, we are fundamentally different in: methods, and tasks. In fact, CGAP is mostly similar to LADS instead of our model.
>
> We only share the similarity with CGAP on using domain prompts to perform augmentation. But the key is how to use them, i.e., augmentation strategies and methods. We significantly differ from CGAP in this sense, which are the major contributions of this paper, leading to constantly better performance on both LADS settings and our setting with much less time cost. We summarize the major differences in the table below:
>
> | Model       | Augmentation Stratagy (Stage-1)                        | Augmentation Method (Stage-1)                                | Domain Prompt                                                | Training Stage Data (Stage-2)             | Task             |
> | ----------- | ------------------------------------------------------ | ------------------------------------------------------------ | ------------------------------------------------------------ | ----------------------------------------- | ---------------- |
> | CGAP        | Train multiple augmentation functions                  | Align global direction with trainable augmentation functions | A given pre-defined list                                     | Augmented and original features           | Object Detection |
> | LADS        | Train multiple augmentation functions                  | Align global direction with trainable augmentation functions | A given pre-defined list (exactlly match test domains)       | Augmented and original features           | Classification   |
> | TEAM (Ours) | *Training-free (calculated with Eq. 9, nonparametric)* | *Align modality direction with nonparametric training-free method* | *(1) Domain-invariant representations (2) a given pre-defined list* | *Domain- invariant and original features* | *Classification* |
>
> In fact, CGAP is mostly similar to LADS in most columns, except for the “task”, while it is different from us in all above aspects. Specifically, CGAP still follows the pipeline of LADS: they first perform optimization for trainable augmentations {$A_j$} (Eq. 3) during the first stage, which are later used for augmenting source images during the training stage. And they all align “global directions” during the training of augmentation functions.
>
> In the following, we would like to clarify each of your concerns:
>
> 1. *“Both papers share fundamental observations: that various domains of the same object differ by constant vectors”*
>
> In fact, the observation of CGAP and Ours is fundamentally **different**. As we stated in the “Augmentation Method” in the above table: CGAP aligns the **global direction** (Eq. 21), following LADS, Styleclip [3], Stylegan-nada [4]. Instead, we first propose to align the **modality direction (Eq. 3)**, motivated by Liang et al [2]. The difference of the two directions, which may seem not very significant at first glance, can lead to notable differences. We demonstrate the difference is critical in detail in Section 4.1: ours are better with (a) more solid theoretical support, (b) better preservation of class information, and (c) milder assumption for an analytical solution. These benefits finally lead to the success of our training-free augmentation. Ablation studies (Tab. 3) also validate this:
>
> | Augmentation Method                 | Average   | ID        | OOD       |
> | ----------------------------------- | --------- | --------- | --------- |
> | LADS                                | 74.99     | 85.33     | 64.85     |
> | Training-free + Align Global Dir.   | 74.67     | 85.21     | 64.12     |
> | Training-free + Align Modality Dir. | **77.16** | **86.61** | **67.71** |
>
> Moreover, as shown in the “Domain Prompt” column in the first table, we further propose to perform a domain-invariant augmentation in Section 4.2 and Fig. 3, which produces domain-invariant features and further reduces the training time (Stage-2 time in Tab.2).

---

> ### Author Response · Authors · 2023-11-16
> **Response to Reviewer GcYV (Part 2)**
>
> 2. *"Additionally, they both tackle the challenge of single-domain generalization. The evaluation contexts differ slightly: CGAP explores weather attributes, while this paper focuses on DomainNet/CUB."*
>
> Yes, we work on single-domain generalization, but the context is different: CGAP focuses on object detection, while we work on classification. We argue that this difference is not trivial. CGAP and LADS train augmentation functions by aligning the global direction, which in fact **originates from image generation research [3,4]**. In the first paragraph of Section 4.1, we question if such direct adaptation fits well for the classification task. Motivated by this, we do not follow them and demonstrate multiple benefits of aligning modality direction with theoretical guarantees, one of which is that it better preserves class information in classification. It is one of the reasons we achieve better results. In other words, we specially designed our methods in the context of classification, and it is not a simple evaluation difference. CGAP still follows existing works (LADS, Styleclip [3], Stylegan-nada [4) to use the global direction.
>
> 3. *"The methodological resemblance is evident: CGAP performs semantic augmentations by computing relative differences in original and target test domains of text embeddings, using these as pseudo targets to generate visual features of the target domains."*
>
> As we stated in the first reply, CGAP is the optimization-based augmentation with “global direction” (Eq. 21), which is adopted by LADS, Styleclip [3], andStylegan-nada [4]. While we align “modality direction” (Eq. 3) in a training-free strategy. Furthermore, we propose to perform a domain-invariant augmentation (also training-free) in Section 4.2 and Fig. 3, which produces domain-invariant augmentations and further reduces the training time (Stage-2 time in Tab. 2).
>
> 4. *"Furthermore, both papers demonstrate that precise domain names are unnecessary for strong performance on unseen domains, as illustrated by CGAP's "random" augmentations in Table 7."*
>
> As we stated in Section 5.4, we carefully check CGAP and find that, it is slightly different from what you mentioned “*as illustrated by CGAP's ‘random’ augmentations*”. In the caption of Tab. 7 in CGAP, the authors claim that, *“While random augmentations are* **worse than no-aug***., clip-random is* **comparable to no-aug***. Only when we give relevant prompts, there is a consistent improvement across datasets.”.* It indicates random/clip-random does not give constant improvements and only their predefined prompts (in Page 5, line 9) can constantly lead to improvements. These predefined prompts are highly relevant to unseen test domains.
>
> By contrast, we use various sets of prompts to study how they can influence our model. In Appendix E and F, we further show results with different sources of domain prompts, which come from different LLMs, including ChatGPT, GPT4, Bard, and New Bing. Besides, we also use the existing prompt template set “ImageNet Templates”, which contains 80 different prompts. The studies validate that our model is robust across different sets of prompts. We also kindly remind that, it is just an additional interesting finding as a supplementary of our key contributions on augmentation strategies and methods.
>
> **Summary for Q1**: In summary, we have three contributions as mentioned in the paper: (1) propose the text-driven domain generalization problem; (2) propose a training-free augmentation method via modality direction alignment with theoretical proofs; and (3) build a framework with our augmentation method that performs domain-invariant augmentations.
>
> We partially agree that (1) may share some similarity in problem definition with CGAP, although CGAP works on a different detection task. However, (2) and (3) are our key contributions that are fundamentally different from CGAP and LADS. We sincerely hope you could take them into consideration.

---

> ### Author Response · Authors · 2023-11-16
> **Response to Reviewer GcYV (Part 3)**
>
> ## Q2. Results
>
> As we mentioned in the first paragraph in Section 5.4, Tab. 1 shows the results under the LADS setting, where exact test domain names are given and LADS is expected to perform well. Tab. 2 is our setting and the interesting part, where exact test domain names are not given. Without exact test domains, the performance gap between LADS and ours is more obvious: **2.86% (OOD)** and **2.17% (AVG)** in CUB-Paintings, and **2.15% (OOD)** and **2.05% (AVG) in** DomainNet. Note that we achieve better results without any augmentation networks nor training during stage-1, while LADS requires training an augmentation network for each given domain prompt. We are significantly more efficient. Besides, our method requires less time in the stage-2 training (four times faster with “invar.” mode in tab. 2). Our model is also more robust across different sets of domain prompts (Appendix E and F show more experiments and details), while LADS only works well when the exact test domain names are given, which is impractical. *We kindly remind that, as mentioned in the paper, out-of-domain (OOD) performance is our major metric as we focus on the domain generalization task, where test domain accuracy is the major metric, like existing works [1,5,6]. To summarize, our model achieves obvious results on our practical setting (Tab. 2) over computing baselines (their performances are very close to each other while we achieve a more obvious improvement)* **without** *training during stage-1 and less time during stage-2*  *with “invar.” mode:*
>
> | Method  | OOD (CUB-Paintings) | OOD (DomainNet) | Average (CUB-Paintings) | Average (DomainNet) |
> | ------- | ------------------- | --------------- | ----------------------- | ------------------- |
> | WiSE-LP | 64.80               | 93.68           | 73.27                   | 94.44               |
> | LADS    | 64.85               | 94.65           | 74.99                   | 94.97               |
> | Ours    | **67.71**           | **96.70**       | **77.16**               | **96.18**           |
>
> Efficiency is also one of our key improvements.
>
> **Reference**
>
> [1] Vidit, Vidit, Martin Engilberge, and Mathieu Salzmann. "CLIP the Gap: A Single Domain Generalization Approach for Object Detection." CVPR, 2023.
>
> [2] Liang et al., Mind the gap: Understanding the modality gap in multi-modal contrastive representation learning. NeurIPS 2022
>
> [3] Patashnik et al., Styleclip: Text-driven manipulation of stylegan imagery, ICCV 2021
>
> [4] Gal et al., “Stylegan-nada: Clip-guided domain adaptation of image generators,” ACM Transactions on Graphics (TOG)
>
> [5] Li et al., A Simple Feature Augmentation for Domain Generalization, ICCV 2021
>
> [6] Lisa et al., Using Language to Extend to Unseen Domains, ICLR 2023

---

> ### Author Response · Authors · 2023-11-21
> **Follow up**
>
> Thanks again for your constructive comments. Would you mind checking our responses to see if your previous concerns/questions have been addressed?
>
> We are also more than happy to provide additional experiments and discussions if you have further questions.

---

> ### Author Response · Authors · 2023-11-22
> **Follow up 2**
>
> Dear reviewer, thank you again for reviewing this paper. As the discussion period is coming to an end, could you please kindly check our responses to see if your concerns are solved?
>
> We sincerely look forward to your response!

---

### Official Review · Reviewer_spPq · 2023-11-01

**Soundness:** 2 fair
**Presentation:** 2 fair
**Contribution:** 2 fair
**Rating:** 5
**Confidence:** 4

**Summary:**

The paper tackles the domain generalization without using the image nor text description of the target domain. The authors propose to use the property of modality gap of CLIP model. Specifically, they augment the embeddings of images in the source domain to the domain-invariant embedding without training. After augmentation, they train the linear probe on the mixture of domain-invariant embedding and original embedding. In inference, they apply the trained linear probe to the embedding of the target image. In the experiment, the authors test their proposed method with and without the text description of the target domain.

**Strengths:**

1. The paper adopts the recent finding of the modality gap in the CLIP embedding space and propose a training-free augmentation which is training-efficient.
2. The paper considers a practical setting where the text description is not available for the target domain.

**Weaknesses:**

1. In **Training-free Augmentation with Modality Direction** (on page 5) the proof of z exists in the CLIP's normalized embedding space is trivial to get.

2. In Table 1 and Table 2, the performance of the proposed method on CUB and DomainNet does not stand out from the baselines with a significant margin.

3. Ablation Studies are limited.

**Questions:**

1. Why can the linear probe trained on the domain-invariant feature and original feature be directly applied to the target feature (without any augmentation)?

2. In Stage 2, why not only train the classifier only on the augmented embeddings?

---

> ### Author Response · Authors · 2023-11-16
> **Response to Reviewer spPq (Part 1)**
>
> We sincerely thank you for your review and helpful comments!
>
> ### W1
> We agree that obtaining a solution for $z$ is not hard mathematically.
> Actually, our intention is not to showcase our skills of solving the equations, but rather to explore the conditions under which a valid solution is available in the normalized embedding space (i.e., training-free feasibility) (Eq. 17 in Appendix B), and its relationship with the pre-trained vision-language (VL) model (Lemma 1).
> We are more interested in $\lambda_{1}$ in Eq. 17 (Appendix B), which determines if the solved $z$ is a valid solution in the embedding space.
> The exploration is critical because it theoretically demonstrates that, if aligning with the global direction (Eq. 21), the condition for the existence of a valid $z$ is more stringent, requiring the cosine similarity of any negative image-text pair is always less than that of the positive pair. This implies that the pre-trained VL model should be perfect (Appendix C). On the other hand, if aligning with our modality direction, the condition for the existence of $z$ is more relaxed: once the embeddings are clustered per modality in the VL model’s embedding space, a solution is readily available. Fortunately, (Liang et al.) [1] validate that the condition is generally satisfied with a VL model trained with a contrastive loss.
> Our ablation studies in Tab. 3 also validate this point: aligning modality direction proves to be more effective:
>
> | Augmentation Method                 | Average   | ID        | OOD       |
> | ----------------------------------- | --------- | --------- | --------- |
> | LADS                                | 74.99     | 85.33     | 64.85     |
> | Training-free + Align Global Dir.   | 74.67     | 85.21     | 64.12     |
> | Training-free + Align Modality Dir. | **77.16** | **86.61** | **67.71** |
>
> ### W2
>
> Tab. 1 presents our results under the LADS setting, where the **exact** test domain names are given. The main results are presented in Tab. 2 in our text-driven domain generalization setting, which is of particular interest to us, where exact test domain names are not given.
>
> It is important to note that it is not an easy task. Under our setting, most competing CLIP-based baselines including WiSE-LP and LADS exhibit very close performance, making further improvements challenging. However, we continue to distinguish ourselves from them (Tab. 2) in AVG and OOD scores:
>
> | Method  | OOD (CUB-Paintings) | OOD (DomainNet) | Average (CUB-Paintings) | Average (DomainNet) |
> | ------- | ------------------- | --------------- | ----------------------- | ------------------- |
> | WiSE-LP | 64.80               | 93.68           | 73.27                   | 94.44               |
> | LADS    | 64.85               | 94.65           | 74.99                   | 94.97               |
> | Ours    | **67.71**           | **96.70**       | **77.16**               | **96.18**           |
>
> Notably, this distinction is achieved **without** the need for any parametric augmentation functions and **without** any training in the first phase. Our work contributes significantly to time efficiency, a crucial aspect beyond accuracy improvement. *We kindly remind that, as mentioned in the paper, out-of-domain (OOD) performance is our major metric as we focus on the domain generalization task, where test domain accuracy is the major metric, like existing works [2,3,4]. To summarize, our model achieves obvious results on our practical setting (Tab. 2) over computing baselines (their performances are very close to each other while we achieve a more obvious improvement)* **without** *training during stage-1 and less time during stage-2*  *with “invar.” mode.
>
> ### W3
>
> We have multiple ablation studies on: (1) augmentation modes and (2) invariant modes in Tab. 3 to show the effectiveness of our augmentation method. We also have ablations on choices of backbone vision-language models in Tab. 11, and on different sources of prompt templates in Tab. 10 and Tab. 11, where we use pre-defined templates such as ImageNet Templates, as well as prompts from different LLMs (e.g., ChatGPT, CPT4, New Bing, and Bard), to demonstrate our robustness across different sets of prompts.
>
> Our existing ablation studies answer most interesting questions. If you believe there is any particular study that we’re missing, we’re more than happy to include.
>
> We add one table regarding your two questions below. We train our model on DomainNet with: ① a mix of augmented data and original data. ② augmented data. ③ original data.
>
>
> | Model                            | Average   | ID        | OOD       |
> | -------------------------------- | --------- | --------- | --------- |
> | ① original data + augmented data | **96.18** | **95.61** | **96.70** |
> | ② augmented data only            | 95.60     | 95.21     | 95.99     |
> | ③ original data only             | 94.39     | 95.03     | 93.75     |
>
> We will discuss it in detail in the response to your questions below.

---

> ### Author Response · Authors · 2023-11-16
> **Response to Reviewer spPq (Part 2)**
>
> ### Questions (Q1 and Q2)
>
> The two questions can be answered by a single statement: The two questions you mentioned are in fact standard pipelines for data augmentation based domain generalization methods [2,3,4], which are also adopted by LADS [4]. As our TEAM model is also a data augmentation based method, we simply follow existing works.
>
> We explain it in detail below. Let's quickly recap our methodology first: We have two model versions: TEAM-full and TEAM-invariant. In TEAM-full, $k$ crafted domain names are provided. We utilize a training-free method to obtain augmented features for $k$ provided domains, which are then **mixed** with the source domain for training on (k+1) domains during stage-2. The evaluation is conducted directly on test domains images without prior image augmentation to a specific domain. **This follows the standard approach in data augmentation based domain generalization (DG) methods [2,3,4], which LADS also follows:**
>
> **(Point. 1 / Q1) Trained model is applied to test domain data without any augmentation.**
>
> **(Point. 2 / Q2) Augmented data is mixed with original data for training.**
>
> In the "invariant" version, augmentation is not performed for $k$ domains. Instead, augmentation is performed for a single "invariant" domain. **The key is, it is still a data augmentation method**, which means the obtained invariant embeddings are still “augmented” data, just like any augmented data for $k$ domains in TEAM-full. The difference from TEAM-full is that, here we set $k$ = 1 (i.e., our so-called "invariant domain").
>
> Therefore, the training and evaluation process are the same with TEAM-full and LADS, i.e., the two bolded points we mentioned above.
>
> In addition, we have done ablation studies on DomainNet regarding your question: we train our model on DomainNet with: ① a mix of augmented data and original data. ② augmented data only. ③ original data only.
>
> | Model                            | Average   | ID        | OOD       |
> | -------------------------------- | --------- | --------- | --------- |
> | ① original data + augmented data | **96.18** | **95.61** | **96.70** |
> | ② augmented data only            | 95.60     | 95.21     | 95.99     |
> | ③ original data only             | 94.39     | 95.03     | 93.75     |
>
> **Reference**
>
> [1] Liang et al., Mind the gap: Understanding the modality gap in multi-modal contrastive representation learning. NeurIPS 2022
>
> [2] Li et al., A Simple Feature Augmentation for Domain Generalization, ICCV 2021
>
> [3] Vidit, Vidit, Martin Engilberge, and Mathieu Salzmann. "CLIP the Gap: A Single Domain Generalization Approach for Object Detection." CVPR, 2023.
>
> [4] Lisa et al., Using Language to Extend to Unseen Domains, ICLR 2023

---

> ### Author Response · Authors · 2023-11-21
> **Follow up**
>
> Thanks again for your constructive comments. Would you mind checking our responses to see if your previous concerns/questions have been addressed?
>
> We are also more than happy to provide additional experiments and discussions if you have further questions.

---

> ### Author Response · Authors · 2023-11-22
> **Follow up 2**
>
> Dear reviewer, thank you again for reviewing this paper. As the discussion period is coming to an end, could you please kindly check our responses to see if your concerns are solved?
>
> We sincerely look forward to your response!

---

### Official Review · Reviewer_mv6P · 2023-11-01

**Soundness:** 3 good
**Presentation:** 3 good
**Contribution:** 3 good
**Rating:** 8
**Confidence:** 3

**Summary:**

The paper presents a novel approach for domain adaptation with the idea of the augment of data with only text descriptions. It utilizes a pre-trained vision-language model to acquire domain-invariant augmentations using text descriptions of arbitrary, unseen domains, even if they don't match the test domains. This method outperforms existing approaches, offering greater efficiency and stronger theoretical support while eliminating the need for predefined text descriptions of all test domains.

**Strengths:**

Good Quality:
The submission is technically sound, in my opinion, and the advantages and limitations of this work are carefully and honestly discussed.

Good Clarity:
The submission is written with clear definitions and formulas. The organization is well-designed.

Good Motivation:
The method is well-motivated, given the recent conclusions from Liang et al. and Zhang et al.

Solid Solution:
By observing the beneficial properties of the multimodal embedding space, the authors successfully extend the findings in Liang et al. and Zhang et al.  to address the domain generalization problem, providing a new method from a novel perspective.

**Weaknesses:**

1.	The authors might consider evaluating their proposed method on a larger dataset, such as WILDS, which provides extensive and diverse real-world data from various domains.

2.	Could you please explain the meaning of "1x" for Time in several tables?

3.	While the authors have successfully developed a method based on the findings of Liang et al. and Zhang et al., the paper itself lacks a sufficient explanation of the embedding space in the context of domain generalization. Although this might not be a critical flaw in the paper, providing theoretical or empirical analyses in this regard could enhance its overall impact and inspiration.

**Questions:**

As the authors mentioned in Figure 2 (right), there might be cases where equation 4 is not satisfied. I am curious if the CLIP features always meet the conditions of equation 4. Could the authors conduct empirical analysis to offer an intuitive understanding of when it fails?

---

> ### Author Response · Authors · 2023-11-16
> **Response to Reviewer mv6P (Part 1)**
>
> Thank you for your time and effort for this review. We appreciate your recognition and constructive questions about our work!
>
> ### W1
>
> Thanks for this insightful suggestion, which points out a critical future direction for this line of research. As we also discussed in Section 6: Limitations, WILDS (Koh et al.) remains a challenge. As our text-driven augmentation requires text descriptions of target domains, it is particularly challenging for complicated natural domain shifts that are hard to capture and express solely through language. We are interested in this problem and will continue to explore it.
>
> ### W2
>
> Sure! Sorry for the confusion of our symbols. "$n$ x" indicates that the stage-2 training time of a particular model is $n$ times that of the LADS baseline. In other words, we consider the training time of LADS as the reference unit time.
>
> For instance, in the following example:
>
> | Model         | Average | ID    | OOD   | Training-free (Stage-1) | Time (Stage-2) |
> | ------------- | ------- | ----- | ----- | ----------------------- | -------------- |
> | LADS          | 74.99   | 85.33 | 64.85 | ✗                       | 1x             |
> | TEAM-*invar.* | 74.67   | 85.21 | 64.12 | ✓                       | 0.23x          |
>
> When the model is TEAM-*invar.*, we have $0.23$x because the training time of TEAM-*invar.* is 0.23 times that of LADS. Therefore, when the model is LADS, we have $1$x because the training time of LADS is the same as that of itself.

---

> ### Author Response · Authors · 2023-11-16
> **Response to Reviewer mv6P (Part 2)**
>
> ### W3
>
> Thanks for the valuable advice. We are also interested to see the geometry of clip embeddings space from the perspective of the training-free augmentation in our domain generalization context. More specifically, we would like to see if the assumptions in Appendix A hold well. We show the following modality gap geometry metrics.
>
> The distance between the modality clusters in the CLIP embedding space is referred to as the modality gap. The instance-level modality gap $g$ is defined as the difference between image and text embeddings for a single pair. The class-level modality gap is defined as $g_{c}$, where for a given class $c$, we calculate the expectation of $g$ over all instances of class $c$ **across different domains**. Finally, we calculate the domain-level modality gap $g_{d}$, where given a specific domain, we calculate the expectation of $g$ over all instances of domain $d$ **across different classes**.
>
> Formally, given an instance pair ($x$, $y$), $g = x - y$, where $x$ and $y$ are CLIP features of an image and a text. We also have:
>
> $g_{c} = x_c - y_c$, where $x_c = E_d[ x_c^d ]$ and  $y_c = E_d[ y_c^d ]$.  Note that $E_d[ x_c^d ]$ means the expectation of all instances of a given class $c$ across all **domains**.
>
> and
>
> $g_{d} = x_d - y_d$, where $x_d = E_c[ x_d^c ]$ and  $y_d = E_c[ y_d^c ]$.  Note that $E_c[ x_d^c ]$ means the expectation of all instances of a given domain $d$ across all **classes**.
>
> We hope that the modality gap $g_{c} $ grants the assumptions we have in Appendix A. Intuitively, if satisfied, given data from one source domain, we can perform our training-free augmentation as  $g_{c} $ is valid across different domains.
>
> More specifically, the two assumptions are:
>
> 1. *The modality gap between corresponding image and text embeddings can be approximated by a constant vector, particularly given a class across different domains. This is verified by computing distributions over ||$g$|| (magnitude) and $cos(g, E_g[g])$ (direction).*
>
> 2. *The modality gap is approximately orthogonal to the span of image embeddings and text embeddings, and image embeddings and text embeddings have a nearly zero mean in the subspace orthogonal to the modality gap. This is verified by computing distributions over $cos(x - E_x[x], E_g[g])$ (orthogonality) and $E_x[x - x^T g’g’]_i$ (center), where $g’ = E_g[g] / || E_g[g] ||$ and $i \in [d]$. The subscript $i$ denotes indexing the $i$-th dimension of the vector.*
>
> The results are listed:
>
> | VL Model - Dataset   | Magnitude - *Instance* | Magnitude - *Class* | Magnitude - *Domain* | Direction - *Instance* | Direction - *Class* | Direction - *Domain* | Orthogonality     | Center           |
> | -------------------- | ---------------------- | ------------------- | -------------------- | ---------------------- | ------------------- | -------------------- | ----------------- | ---------------- |
> | CLIP - DomainNet     | 1.32 $\pm$ 0.034       | 1.19 $\pm$ 0.027    | 1.32 $\pm$ 0.029     | 0.76 $\pm$ 0.051       | 0.89 $\pm$ 0.023    | 0.77 $\pm$ 0.049     | 0.01 $\pm$  0.107 | 0.00 $\pm$ 0.024 |
> | CLIP - CUB Paintings | 1.29 $\pm$ 0.035       | 1.18 $\pm$ 0.029    | 1.30 $\pm$ 0.038     | 0.70 $\pm$ 0.051       | 0.88 $\pm$ 0.029    | 0.74 $\pm$ 0.045     | 0.00 $\pm$  0.095 | 0.00 $\pm$ 0.026 |
>
> We can observe that, the orthogonality and center are all close to 0 with small STD, which means assumption 2 is well satisfied. For magnitude and direction, the class-level modality gap (**across domains**) better satisfies assumption 1, as we have higher scores for direction on both datasets. It satisfies our expectation that, the class-level modality gap $g_{c}$ better grants the assumptions. It indicates that, given data from one source domain, we can perform our training-free augmentation because the modality gap $g_{c} $ of a given class $c$ is valid across different domains.

---

> ### Author Response · Authors · 2023-11-16
> **Response to Reviewer mv6P (Part 3)**
>
> ### Question
>
> It is an interesting question. Let's illustrate it with an example, where the source domain is "painting" and target domain is "photo". Given an image-text pair from source data, say, (apple_painting.jpg, "a painting of an apple"), and a text description from target domain: "a photo of an apple", intuitively, equation 4 means: in the CLIP embedding space, "a painting of an apple" is closer to  "a photo of an apple"  than to apple_painting.jpg. In other words, embeddings of the **same** modality should be closer to each other than embeddings of the same semantic meaning but different modality. This is what we expect.
>
> Fortunately, Liang et al. reveal that, when a vision-language model is pre-trained with a contrastive loss, the embeddings are clustered per modality, which naturally guarantees what we expect. **And in practice, we find it always holds.** It is reasonable because the CLIP model is well-trained with 400M data, during which time the embeddings are very well clustered per modality.
>
> You may refer to Fig. 1b (https://arxiv.org/pdf/2203.02053.pdf). Intuitively, if there is a failure case, you could imagine a blue point appearing among the red points, which indicates a text embedding is closer to one or some image embeddings than other text embeddings. In our practice with the pre-trained CLIP model, it never happens.
>
> Liang et al. further explain why they are so separated from each other from three aspects: cone effect, initializations, and contrastive learning objective.

---

> > ### Comment · Reviewer_mv6P · 2023-12-05
> >
> > Thanks for the authors' feedback. It addresses all my concerns. I would maintain my original score.

---

### Meta-Review · Area_Chair_Y2gw · 2023-12-17

**Metareview:**

This paper received borderline and mixed reviews - there was some appreciation for the simplicity of the method and exposition. But there were concerns about the significance and novelty of the results. Upon careful consideration of all the points raised, and also a personal review of the paper, here are some weaknesses that should be addressed before being ready for publication:

(1) The primary motivation of the paper: "training free" vs "w/o oracle". I think the title, abstract and intro emphasize the "without oracle" aspect, but I think this is not that significant a problem. It is not clear that getting text descriptions can be cumbersome, especially in the datasets that are evaluated in the paper. They are all datasets where text descriptions could be obtained with virtually no effort (like in LADS). So the training-free could be better emphasized right from the beginning.

(2) Regarding the "training free" aspect, the current exposition is quite confusing. There are a lot of loss functions and details in the paper but also some important claims moved to the appendix. I recommend the authors restructure the description to highlight from first principles what a training free approach would entail and how to go about developing new methods there. That would also help clarify the novelty concern raised by some reviewers. Also, is it really that important to skip the training, or can one learn a single data augmentation network on all the augmented images (unlike LADS that has a separate augmentation network)? What are the cost-benefit tradeoffs here?

(3) More extensive datasets and comparisons: would be good to consider more complex datasets (even if just to udnerstand the boundaries of the methods).

**Justification For Why Not Higher Score:**

Some concerns around significance, novelty and clarity. I think the paper is borderline on all aspects and I don't see any particular aspect that helps push this paper to accept.

**Justification For Why Not Lower Score:**

N/A

---

### Decision · Program_Chairs · 2024-01-16

Reject